# Rank $Q$ E-string on a torus with flux

**Sara Pasquetti[1]⋆, Shlomo S. Razamat[2], Matteo Sacchi[1] and Gabi Zafrir[3]**

**1** Dipartimento di Fisica, Università di Milano-Bicocca & INFN,
Sezione di Milano-Bicocca, I-20126 Milano, Italy
**2** Department of Physics, Technion, Haifa, 32000, Israel
**3** Kavli IPMU (WPI), University of Tokyo, Kashiwa, Chiba 277-8583, Japan

⋆ sara.pasquetti@gmail.com

## Abstract

We discuss compactifications of rank $Q$ E-string theory on a torus with fluxes for abelian subgroups of the $E_8$ global symmetry of the $6d$ SCFT. We argue that the theories corresponding to such tori are built from a simple model we denote as $E[USp(2Q)]$. This model has a variety of non trivial properties. In particular the global symmetry is $USp(2Q) \times USp(2Q) \times U(1)^2$ with one of the two $USp(2Q)$ symmetries emerging in the IR as an enhancement of an $SU(2)^Q$ symmetry of the UV Lagrangian. The $E[USp(2Q)]$ model after dimensional reduction to $3d$ and a subsequent Coulomb branch flow is closely related to the familiar $3d$ $T[SU(Q)]$ theory, the model residing on an S-duality domain wall of $4d$ $\mathcal{N} = 4$ $SU(Q)$ SYM. Gluing the $E[USp(2Q)]$ models by gauging the $USp(2Q)$ symmetries with proper admixtures of chiral superfields gives rise to systematic constructions of many examples of $4d$ theories with emergent IR symmetries. We support our claims by various checks involving computations of anomalies and supersymmetric partition functions. Many of the needed identities satisfied by the supersymmetric indices follow directly from recent mathematical results obtained by E. Rains.



# 1 Introduction

Quantum field theories in various space-time dimensions are interconnected by a variety of relations. These for example include RG flows, dimensional reductions, and dualities. For supersymmetric theories such interconnections can be effectively probed using robust quantities such as anomalies and supersymmetric partition functions. Deeper understanding of the relations between models often leads to novel physical insights. Examples of these include deducing dualities from compactifications, emergence of symmetry in the IR, novel constructions of CFTs, as well as interesting interrelations between a priori unrelated subjects in mathematical physics. In this paper we will discuss an example of an interconnection between different constructions of certain quantum field theories which has all of the features mentioned above.

Concretely we will *first* construct a family of four dimensional $\mathcal{N} = 1$ theories corresponding to the compactification of a six dimensional SCFT, the rank $Q$ E-string theory, on a torus with fluxes for $E_8$ subgroup of its $E_8 \times SU(2)_L$ symmetry group. This construction is a generalization of the results for $Q = 1$ obtained in [1] which follows the general ideas of relating four dimensional models to compactifications of six dimensional theories initiated in [2, 3] and pursued in various setups, see *e.g.* [4–21]. The geometric construction starting from six dimension allows us to make predictions regarding the global symmetries and the anomalies of the four dimensional models.

As in [1] we will build the four dimensional models by combining together tube theories corresponding to compactifications on two punctured spheres with flux. In the $Q = 1$ case the basic tube theory consists of an $SU(2) \times SU(2)$ bifundamental with two octets of fundamentals and an additional singlet. In the higher rank case the basic tube theory will be constructed from the basic building block theory, which we will denote by $E[USp(2Q)]$, depicted in Figure 1. The $E[USp(2Q)]$ theory in the UV has $SU(2)^Q \times USp(2Q) \times U(1)^2$ global symmetry which we will argue enhances to $USp(2Q) \times USp(2Q) \times U(1)^2$ in the IR.

The basic tube theory will then be defined by connecting two octets of fundamentals to the $E[USp(2Q)]$ theory as depicted in Figure 2. The theories obtained by gluing these higher rank tubes will perfectly match the predictions obtained via the six dimensional construction.

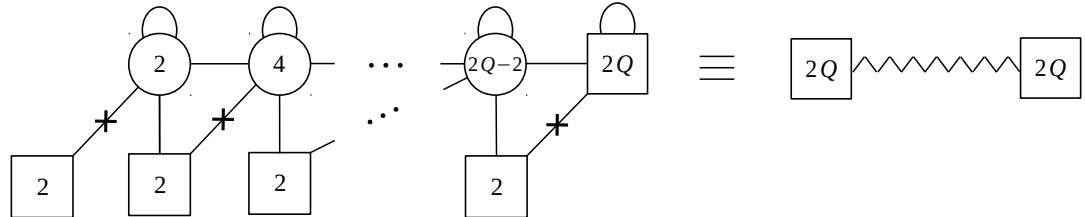

Figure 1: The $E[USp(2Q)]$ quiver theory. Each gauge node has $USp(2n)$ symmetry, while each flavor node carries $SU(2)$ symmetry. The crosses indicate singlets flipping the diagonal mesons. The lines starting and ending on the same node stand for two-index antisymmetric field which for the $USp(2Q)$ flavor node is also traceless. The IR global symmetry group is $USp(2Q) \times USp(2Q) \times U(1)^2$. On the rhs we introduce a compact representation of the IR SCFT to which $E[USp(2Q)]$ flows to.

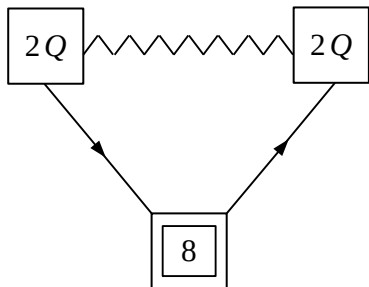

Figure 2: The basic tube theory: we couple the $E[USp(2Q)]$ block to two sets of chiral fields in the fundamental and anti-fundamental representation of $SU(8)$ respectively. We use simple circles and squares to denote gauge and flavor $USp(2n)$ nodes and double-line squares to denote flavor $SU(k)$ nodes.

In particular, they will display the predicted enhanced symmetries, such as $SU(2) \times E_7 \times U(1)$ or $SU(2) \times SO(14) \times U(1)$ depending on the flux chosen in the compactification.

Thus we see a *second* interesting phenomenon: the geometric construction leads to IR symmetry enhancement properties that the theories obtained by gluing the basic tubes need to satisfy. As the symmetry of the four dimensional model is determined by the six dimensional symmetry and details of the compactification, such as flux, though the tube building block has lower symmetry the combined models might have larger symmetry. Also, combining the tube building blocks in different orders does not affect the IR fixed point which leads to various IR dualities. For both such effects see *e.g.* [16, 17].

The four dimensional theories corresponding to torus compactifications will be built gluing tube theories by gauging both $USp(2Q)$ symmetries associated to the two punctures. In particular, as an example of a *third* interesting effect, this means that in order to construct these models we need to gauge a symmetry which only appears in the IR and is not visible in the UV. This is an example of a novel construction of QFTs which played an important role in various setups in recent years. In particular this idea was applied to construct a Lagrangian for the $E_6$ Minahan-Nemeschansky SCFT [22] and later also the $E_7$ SCFT [23], as well as a variety of Lagrangians of other strongly coupled SCFTs [7, 18].

*Fourth*, by dimensionally reducing the $E[USp(2Q)]$ theory farther to three dimensions and

studying its flows under different real mass deformations we will show it reduces to various known theories. In particular we can reach the $\mathcal{N} = 2$ $FM[SU(Q)]$ theory recently discussed in [24] and with a further flow the $\mathcal{N} = 2$ $FT[SU(Q)]$ discussed in [25, 26], which up to an extra set of singlets fields coincides with the widely studied $\mathcal{N} = 4$ $T[SU(Q)]$ [27]. All these three dimensional theories display, as their four dimensional ancestor $E[USp(2Q)]$ theory, a non-trivial IR symmetry enhancement and interesting self-duality properties.

*Fifth*, the whole construction is tightly tied to seemingly unrelated topics in mathematical physics. The integral form of the superconformal index [28–30] (see for a review [31]) of the $E[USp(2Q)]$ theory coincides with the interpolation kernel recently discussed by E. Rains [32] as analytic continuation of the elliptic interpolation functions. The interpolation kernel satisfies various remarkable properties inherited from its definition in terms of interpolation functions and here we reinterpret these properties as dualities for the $E[USp(2Q)]$ theory which play a key role in unraveling the network of relations presented in this paper. Integral identities encoding supersymmetric dualities at the level of partition functions calculated via localisation (see [33] for a review) have often been discussed independently in the mathematical literature. For example, as noted in [30], the fact that pairs of Seiberg dual theories [34] have the same superconformal index is due to very non-trivial integral identities which were earlier proven in [35, 36]. Indeed there are many interrelations between math and physics literature in this context (see *e.g.* [37–44] for more examples).

There is a further connection. As mentioned above once compactified to three dimension and subject to a real mass deformation the $E[USp(2Q)]$ theory reduces to the $FM[SU(Q)]$ quiver theory. Then, as shown in [24, 45], the $\mathbb{S}^2 \times \mathbb{S}^1$ partition function of the $FM[SU(Q)]$ theory reduces in a suitable limit to another very interesting mathematical object, the kernel function defined in [46]. This kernel function, defined as a Q-dimensional complex integral, plays a key role in the manipulation of complex integrals encoding the free field representation of 2d CFT correlators. In [24, 45] these manipulations were re-interpreted as dualities between three-dimensional supersymmetric gauge theories.

The paper is organized as follows. In section 2 we discuss the six dimensional rank $Q$ E-string theory which is the starting point of our considerations. We review its tubes and tori compactifications and make prediction for the global symmetries and anomalies of the four dimensional models. In section 3 we introduce the $E[USp(2Q)]$ theory and discuss various properties it satisfies. In section 4 we show how the $E[USp(2Q)]$ theory is related to compactifications of the rank $Q$ E-string theory on a torus with fluxes. In section 5 we discuss the reduction of $E[USp(2Q)]$ to three dimensions and the relation of it to $T[SU(Q)]$. We finish in section 6 with several comments about the results. The bulk of the paper is supplemented with appendices discussing various technical facets of the computations.

## 2 Six dimensions

The 6$d$ SCFT, compactifications of which we are about to consider, is the rank $Q$ E-string theory, and we shall begin our discussion by listing several properties of this SCFT that will be useful later. The rank $Q$ E-string SCFT can be engineered in string theory as the theory living on $Q$ M5-branes probing an M9-plane. In addition to the 6$d$ superconformal symmetry, it has an $SU(2)_L \times E_8$ global symmetry. In the brane construction the $E_8$ comes from the gauge symmetry on the M9-plane, and the $SU(2)_L$ comes from the $SO(4) = SU(2)_L \times SU(2)_R$ symmetry acting on the directions of the M9-plane orthogonal to the M5-branes, where the other $SU(2)_R$ is the R-symmetry. The matter spectrum consists of $Q$ tensor multiplets. While it has no known Lagrangian description in 6$d$, its compactification to lower dimensions leads to more approachable theories and we shall consider these.

It is known that when compactified on a finite radius circle to $5d$, with a proper holonomy inside $E_8$,[1] it flows to a $5d$ gauge theory with a $USp(2Q)$ gauge group, an antisymmetric hypermultiplet and eight fundamental hypermultiplets [48]. We can also consider the compactification without the holonomy in the zero radius limit, where the theory flows to a $5d$ SCFT with $SU(2)_L \times E_8$ global symmetry, which was originally found in [47]. We can consider turning back the holonomy, which is mapped to a mass deformation that causes the $5d$ SCFT to flow to a $5d$ gauge theory with gauge group $USp(2Q)$ and matter being an antisymmetric and seven fundamental hypermultiplets. We note that one can continue with circle compactification to get other interesting theories with $E_8$ global symmetry in lower dimensions. For instance the compactification on a torus leads [48] to the rank $Q$ Minahan-Nemeschansky $E_8$ strongly interacting SCFTs [49].

## 2.1 Rank $Q$ E-string compactifications on tori and tubes: $6d$ predictions

We are interested in the compactification of the E-string SCFTs on Riemann surfaces with fluxes in their $E_8$ global symmetry[2]. The majority of the discussion in this section was already worked out in [1], and we shall merely summarize the main parts here. As previously mentioned we are interested in compactifications that have a non-trivial flux in the $U(1)$ subgroups of the $E_8$ global symmetry. To enumerate the fluxes it is convenient to introduce a flux basis. For this we use the $SO(16) \subset E_8$ and parametrize the fluxes via the eight fluxes in the $SO(2)^8 \subset SO(16)$. The fluxes are then given by a vector of eight numbers, $(n_1, n_2, ..., n_8)$. For more information see [1] and appendix A.

The major thing that will concern us here is the determination of the anomalies of the resulting $4d$ theories from the anomalies of the original $6d$ SCFT. The anomalies generally receive two contributions. One is from the integration of the anomaly polynomial of the $6d$ SCFT on the Riemann surface [4], and the other is the contribution from the degrees of freedom associated with the punctures, if these are present. We shall begin by discussing the first contribution and then move on to discuss the second one.

Barring the issue of punctures, the anomalies of the resulting $4d$ theories can be evaluated by integrating the anomaly polynomial eight form of the $6d$ SCFT on the Riemann surface. This calculation was already performed in [1], based on the anomaly polynomial of the rank $Q$ E-string SCFTs that was evaluated in [51], and here we shall just quote the results. We integrate the $6d$ anomaly polynomial on a genus $g$ Riemann surface with $s$ punctures. The flux on the surface is given by the vector $(n_1, n_2, ..., n_8)$ in the eight Cartans of $E_8$, $U(1)_{F_i}$, as introduced previously. The flux is normalized such that $\int F_{U(1)_{F_i}} = 2\pi n_i$. We consider vectors $(n_1, n_2, ..., n_8)$ breaking $E_8 \to U(1) \times G$. We also define $z$ as the flux in the $U(1)$ whose commutant is $G$ and we introduce the coefficient $\xi_G$ parametrising [1] the choice of the $U(1)$ in $E_8$ in which the flux is turned on. For $\xi_G = 1$ the commutant of the $U(1)$ is $E_7$, for $\xi_G = 2$ it is $SO(14)$, for $\xi_G = 3$ it is $E_6 \times SU(2)$, for $\xi_G = 4$ it is $SU(8)$, for $\xi_G = 6$ it is $SU(3) \times SO(10)$, for $\xi_G = 7$ it is $SU(2) \times SU(7)$, for $\xi_G = 10$ it is $SU(4) \times SU(5)$, and for $\xi_G = 15$ it is $SU(2) \times SU(3) \times SU(5)$. The $U(1)_R$ symmetry we use is the descendent of the Cartan of the $6d$ $SU(2)_R$. Its $6d$ origin makes it useful to work with for the purpose of anomaly calculations,

---

[1]That an holonomy is necessary can be seen from the fact that the $E_8$ symmetry is broken in the low-energy gauge theory. More specifically, to get the low-energy $5d$ gauge theory the holonomy must be tuned with the radius, see [47].

[2]Flux compactifications of $6d$ SCFTs to four dimensions were first discussed in [50].

though it is in general not the superconformal R-symmetry. The anomalies are

$$\text{Tr}(U(1)_R^3) = (g-1+\tfrac{s}{2})Q(4Q^2+6Q+3), \quad \text{Tr}(U(1)_R) = -(g-1+\tfrac{s}{2})Q(6Q+5),$$

$$\text{Tr}(U(1)) = -12Qz\xi_G, \qquad \text{Tr}(U(1)^3) = -12Qz\xi_G^2,$$

$$\text{Tr}(U(1)_R U(1)^2) = -2Q(Q+1)(g-1+\tfrac{s}{2})\xi_G, \qquad \text{Tr}(U(1)U(1)_R^2) = 2Q(Q+1)\xi_G z,$$

$$\text{Tr}(U(1)_R SU(2)_L^2) = -\frac{Q(Q^2-1)(g-1+\tfrac{s}{2})}{3}, \qquad \text{Tr}(U(1)SU(2)_L^2) = -\frac{Q(Q-1)}{2}\xi_G z. \tag{1}$$

We can use the above anomalies to write a trial $a$ function and perform a maximization to obtain candidate values for the superconformal $a$ and $c$ anomalies. This always comes with the caveat of having no accidental abelian symmetries, which is not always satisfied. Nevertheless, if we have matched the symmetries between $4d$ and $6d$ the analogous naive computation should produce the same result and thus we quote it here. The anomalies are [1],

$$a = \frac{\sqrt{2\xi_G}Q(3Q+5)^{\frac{3}{2}}}{16}|z|, \qquad c = \frac{\sqrt{2\xi_G(3Q+5)}Q(3Q+7)}{16}|z|. \tag{2}$$

## 2.2 Punctures

We now move on to discussing the contribution of the punctures to the anomalies, where we specifically concentrate on the contribution from the degrees of freedom associated with the punctures rather than the geometric contribution which was previously discussed. The calculation of this contribution of the punctures to the anomalies was set up in [1, 16, 17][3], and here we shall briefly review and apply it to the case at hand. The basic idea is to consider the region around a puncture and deform it so as to look like a long thin tube ending at the puncture. We can then compactify the $6d$ SCFT on the circle of the tube and get the reduced $5d$ theory on an interval ending with the puncture. Particularly, we shall assume that the necessary holonomy as been turned on around the tube so that the reduced $5d$ theory is the IR free $USp(2Q)$ gauge theory with an antisymmetric hyper and eight fundamental hypers that was introduced previously. The puncture then can be described as a boundary condition of this $5d$ gauge theory.

This leads us to consider boundary conditions of $5d$ gauge theories preserving four supercharges. These can be described as giving Dirichlet or Neumann boundary conditions to various multiplets on the boundary. Specifically, close to the boundary the $5d$ bulk fields approach the $4d$ boundary and can be decomposed in terms of $4d$ $\mathcal{N}=1$ superfields. The boundary conditions can then be described as assigning Dirichlet or Neumann boundary conditions to those superfields.

There are in principal many different possible boundary conditions leading to the many different punctures that exist in these types of construction. Here we shall only consider one type, which is the one considered in [1] for $Q=1$, generalized to the case of generic $Q$. This type of puncture can be thought of as a generalization of the so called maximal punctures of class S theories [2]. The boundary conditions associated with this choice are as follows. First we decompose the $5d$ vector multiplet to the $4d$ $\mathcal{N}=1$ vector multiplet and adjoint chiral on the boundary. We then give Dirichlet boundary conditions for the $\mathcal{N}=1$ vector and Neumann boundary conditions for the adjoint chiral. Note that as the vector multiplet is given Dirichlet boundary conditions, the $5d$ $USp(2Q)$ gauge symmetry becomes non-dynamical at the boundary. As a result it becomes a global symmetry associated with the puncture.

---

[3]See [52] for the discussion in case of $(2,0)$ SCFT.

Likewise we can decompose the hypermultiplets to two chiral fields in conjugate representations, and give Dirichlet boundary conditions to one and Neumann boundary conditions to the other. Here we have a choice as to which chiral gets which boundary conditions, and this leads to slightly different punctures. This difference is usually referred to as the sign of the puncture.

We next want to consider the contribution of the degrees of freedom at the boundary to the anomalies. This is known to be given by half the $4d$ anomalies expected from the matter given Neumann boundary conditions, see [1] for the details. We next evaluate these for the punctures considered here. First we consider the anomalies involving the $U(1)_R$ Cartan of the $SU(2)_R$ symmetry. These only receive contributions from the adjoint chiral as the fermions in the hypermultiplet are $SU(2)_R$ singlets. Specifically the fermion in the adjoint chiral has charge $-1$ under $U(1)_R$, is in the adjoint of the $USp(2Q)$ symmetry associated with the puncture, and is a singlet under the other global symmetries. As a result it contribute to the anomalies:

$$\mathrm{Tr}(U(1)_R^3) = -\frac{Q(2Q+1)}{2}, \qquad \mathrm{Tr}(U(1)_R) = -\frac{Q(2Q+1)}{2},$$
$$\mathrm{Tr}(U(1)_R USp(2Q)^2) = -\frac{(Q+1)}{2}. \tag{3}$$

Next we want to consider the anomalies under the $SU(2)_L$ global symmetry. It receives contributions only from the antisymmetric hyper, the two chirals in which form a doublet of this symmetry. As we give different boundary conditions to them, the puncture breaks $SU(2)_L$ to its $U(1)_L$ Cartan, and the anomalies expected for this symmetry are:

$$\mathrm{Tr}(U(1)_L^3) = q^3 \frac{(Q(2Q-1)-1)}{2}, \qquad \mathrm{Tr}(U(1)_L) = q\frac{(Q(2Q-1)-1)}{2},$$
$$\mathrm{Tr}(U(1)_L USp(2Q)^2) = q\frac{(Q-1)}{2}. \tag{4}$$

Here $q$ is the charge under $U(1)_L$ which depends on the normalization and the sign. We will use in what follows normalization of charges such that $q = -\frac{1}{2}$.

Finally the contribution to the anomalies of the $U(1)$ for which we are turning on the flux receives contributions only from an octet of the $USp(2Q)$ fundamental hypers carrying charge $q_a$ with $a = 1, \cdots 8$, and it is given:

$$Tr(U(1)^3) = Q\sum_{a=1}^{8} q_a^3, \qquad Tr(U(1)) = Q\sum_{a=1}^{8} q_a, \qquad Tr(U(1)USp(2Q)^2) = \frac{1}{4}\sum_{a=1}^{8} q_a. \tag{5}$$

We will use a normalization for $U(1)$ such that all the octet fields have the same charge $q_a = -\frac{1}{2}$.

# 3 The $E[USp(2Q)]$ theory

In this section we introduce the $E[USp(2Q)]$ theory. This model satisfies a lot of interesting properties, which will be discussed in detail in this section. In the next section we will also see that it serves as a building block to construct theories obtained by compactifications on a torus with flux in $E_8$ of the rank $Q$ E-string, and in section 5 we will show how upon reduction to three dimensions it is related to the $T[SU(Q)]$ theory.

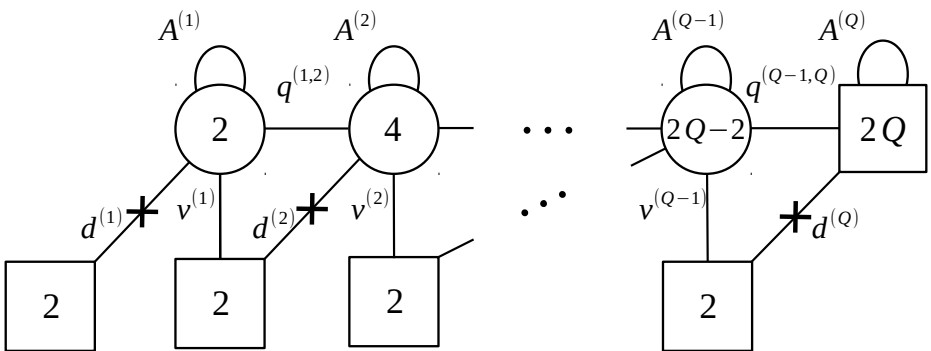

Figure 3: Fields in the $E[USp(2Q)]$ theory. The crosses represent the gauge singlet fields $b_n$.

## 3.1 Symmetry enhancement and dynamics

$E[USp(2Q)]$ is the $4d$ $\mathcal{N}=1$ quiver gauge theory represented in Figure 3. It consists of a $\prod_{n=1}^{Q-1} USp(2n)$ gauge group with several chiral fields in the singlet, fundamental, bifundamental and antisymmetric representation, which we denote as follows:

- $q^{(n,n+1)}$ is a chiral field in the bifundamental representation of $USp(2n) \times USp(2(n+1))$;

- $d^{(n)}$ is a chiral field in the fundamental representation of $USp(2n)$ which is connected to the $n$-th $SU(2)$ flavor node diagonally;

- $v^{(n)}$ is a chiral field in the fundamental representation of $USp(2n)$ which is connected to the $(n+1)$-th $SU(2)$ flavor node vertically;

- $A^{(n)}$ is a chiral field in the antisymmetric representation of $USp(2n)$; for $n = Q$ it is actually a gauge singlet in the traceless[4] antisymmetric representation of the $USp(2Q)_M$ global symmetry, which we will often denote simply by $A_x$;

- $b_n$ is a gauge singlet that is coupled to a gauge singlet built from $d^{(n)}$ through a superpotential which will be discussed momentarily.

We assign R-charge, which we denote as $R_0$, zero to fields $q^{(n,n+1)}$ and $d^{(n)}$, and $R_0$ charge two to fields $b_n$, $A^{(n)}$ and $v^{(n)}$. This is not the superconformal R-symmetry but it is anomaly free and consistent with the superpotentials we will turn on, and it is the simplest one we can write. We will discuss the superconformal R-symmetry momentarily.

---

[4]The tracelessness is in terms of the following trace:

$$\text{Tr}_{2n} A = J_{ij}^{(n)} A^{ji} = A^i{}_i\,,$$

where $J^{(n)}$ is an antisymmetric tensor associated to the $USp(2n)$ group defined as

$$J^{(n)} = \mathbb{I}_n \otimes i\,\sigma_2\,.$$

For $SU(2)$ this is simply the usual $\epsilon_{\alpha\beta}$ tensor. In our conventions, indices that appear both up and down are contracted and we can use the $J^{(n)}$ tensor to raise and lower indices. Thus, in many of the expressions we will write some $J^{(n)}$ tensors are actually implied. For example

$$\text{Tr}_{2n}(AB) = A_{ij} B^{ji} = J_{ik}^{(n)} J_{jl}^{(n)} A^{kl} B^{ji}\,.$$

In order to write the superpotential in a compact form, we introduce for each $USp(2n)$ gauge node the following mesonic fields transforming in its antisymmetric representation:

$$\mathbb{M}^{(n)}_{L,ij} = J^{(n-1)}_{ab} q^{(n-1,n)a}{}_i q^{(n-1,n)b}{}_j = q^{(n-1,n)a}{}_i q^{(n-1,n)}_{aj}$$

$$\mathbb{M}^{(n)ij}_R = J^{(n+1)ab} q^{(n,n+1)i}{}_a q^{(n,n+1)j}{}_b = q^{(n,n+1)i}{}_a q^{(n,n+1)ja} . \tag{6}$$

For the first node $n = 1$ we only have the right meson $\mathbb{M}^{(1)}_R$, while for the last flavor node $n = Q$ we only have the left meson $\mathbb{M}^{(Q)}_L$ which is actually a gauge invariant operator.

The superpotential consists of three parts. The first one is the cubic interaction between the bifundamentals and the antisymmetrics, then we have the cubic interaction between the chirals in each triangle of the quiver and finally the flip terms with the singlets $b_n$ coupled to the diagonal mesons:[5]

$$
\begin{aligned}
\mathcal{W}_{E[USp(2Q)]} = {} & \mathrm{Tr}_2\left(A^{(1)}\mathbb{M}^{(1)}_R\right) + \sum_{n=2}^{Q-1}\mathrm{Tr}_{2n}\left[A^{(n)}\left(\mathbb{M}^{(n)}_R - \mathbb{M}^{(n)}_L\right)\right] - \mathrm{Tr}_{2Q}\left(A_x\mathbb{M}^{(Q)}_L\right) + \\
& + \sum_{n=1}^{Q-1}\mathrm{Tr}_2\,\mathrm{Tr}_{2n}\,\mathrm{Tr}_{2(n+1)}\left(v^{(n)}q^{(n,n+1)}d^{(n+1)}\right) + \sum_{n=1}^{Q} b_n\,\mathrm{Tr}_2\,\mathrm{Tr}_{2n}\left(d^{(n)}d^{(n)}\right) . \tag{7}
\end{aligned}
$$

Notice that in the last term of the first line the antisymmetric $A_x$ is flipping the gauge invariant operator $\mathbb{M}^{(Q)}_L$.

The manifest global symmetry of the theory is

$$USp(2Q)_x \times \prod_{n=1}^{Q} SU(2)_{y_n} \times U(1)_t \times U(1)_c . \tag{8}$$

We claim that the $\prod_{n=1}^{Q} SU(2)_{y_n}$ symmetry of the quiver enhances to $USp(2Q)_y$ at low energies, so that the global symmetry becomes

$$USp(2Q)_x \times USp(2Q)_y \times U(1)_t \times U(1)_c . \tag{9}$$

We will give several pieces of evidence for this enhancement later on. The charges of all the chiral fields under the two $U(1)$ symmetries as well as their trial R-charge are represented in Figure 4.

The abelian symmetries can mix with the R-charge at low energy and the true R-charge at the superconformal fixed point will be

$$R = R_0 + \mathfrak{c}q_c + \mathfrak{t}q_t , \tag{10}$$

where $R_0$ is the trial R-charge, $q_c$ and $q_t$ are the charges under the two $U(1)$ symmetries and $\mathfrak{c}$ and $\mathfrak{t}$ are mixing coefficients that are determined via a-maximization [53].

The fact that we only have two $U(1)$ global symmetries descends from the requirement that the superpotential is uncharged under all the global symmetries and has R-charge 2 and that $U(1)_R$ is non-anomalous[6]. Indeed, if we parametrize $U(1)_t$ and $U(1)_c$ such that the last

---

[5] For $Q = 1$ the last term is

$$b_1\,\mathrm{Tr}_2\,\mathrm{Tr}_2\,d^{(1)}d^{(1)} = b_1\epsilon^{\alpha\beta}\epsilon^{\gamma\delta}d^{(2)}_{\beta\delta}d^{(2)}_{\alpha\gamma} = b_1\det d^{(2)} .$$

[6]This requirement translates into the condition

$$\sum_f T(\mathcal{R}_f)R_f = 0,$$

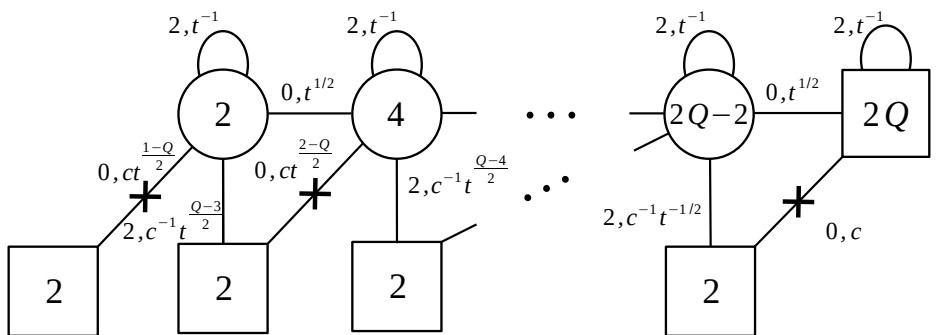

Figure 4: Trial R-charges and charges under $U(1)_c \times U(1)_t$. The $q_c$ and $q_t$ charges are given by the exponents of the fugacities $c$ and $t$.

bifundamental has R-charge $R[q^{(Q-1,Q)}] = \frac{1}{2}\mathfrak{t}$ while the last diagonal has R-charge $R[d^{(Q)}] = \mathfrak{c}$ (as we do in Figure 4), then the superpotential terms that couple the bifundamentals to the antisymmetrics imply that

$$R[\Phi^{(n)}] = 2 - \mathfrak{t}, \qquad R[q^{(n,n+1)}] = \frac{1}{2}\mathfrak{t}, \qquad \forall n. \tag{11}$$

The cubic superpotential associated to the last triangle of the quiver then forces the last vertical to have R-charge

$$R[v^{(Q-1)}] = 2 - \frac{1}{2}\mathfrak{t} - \mathfrak{c}. \tag{12}$$

The R-charge of the next diagonal is instead fixed by the requirement that $U(1)_R$ is non-anomalous at the $USp(2(Q-1))$ node

$$R[d^{(Q-1)}] = -\frac{1}{2}\mathfrak{t} + \mathfrak{c}. \tag{13}$$

Proceeding along the tail in this way we can fix all the R-charges in terms of the mixing coefficient $\mathfrak{t}$ and $\mathfrak{c}$ only. If the $(n+1)$-th diagonal has R-charge

$$R[d^{(n+1)}] = \frac{n+1+Q}{2}\mathfrak{t} + \mathfrak{c}, \tag{14}$$

then the cubic superpotential will fix

$$R[v^{(n)}] = 2 - \frac{n+2-Q}{2}\mathfrak{t} - \mathfrak{c}, \tag{15}$$

where the sum is over all the fermions in the theory which are in the representation $\mathcal{R}_f$ of the gauge group and have R-charge $R_f$. $T(\mathcal{R})$ is one-half the Dynkin index of the representation $\mathcal{R}$ and it is defined as

$$\text{Tr}\left(T_{\mathcal{R}}^a T_{\mathcal{R}}^a\right) = T(\mathcal{R})\delta^{ab}$$

where $T_{\mathcal{R}}^a$ are the generators of the gauge group in the representation $\mathcal{R}$. For our purposes, it will be useful to recall that for $USp(2n)$ we have

$$T(\mathbf{fund.}) = 1/2$$
$$T(\mathbf{adj.}) = n + 1$$
$$T(\mathbf{antisymm.}) = n - 1$$

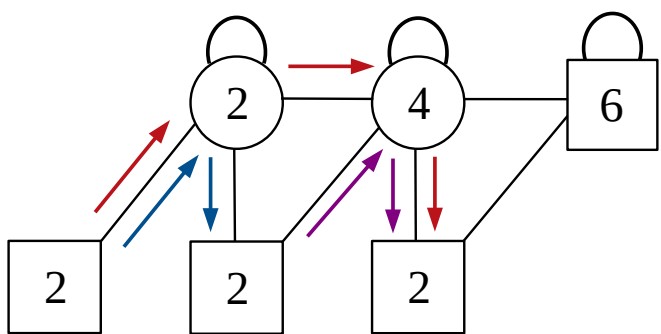

Figure 5: Operators in the upper triangle of the $A_y$ matrix in the $Q = 3$ case.

while the condition that $U(1)_R$ is preserved at the $USp(2n)$ node will imply

$$R[d^{(n)}] = \frac{n+Q}{2}\mathfrak{t} + \mathfrak{c}. \tag{16}$$

Notice that at each step $\mathfrak{c}$ gets shifted by $\mathfrak{c} \to \mathfrak{c} - \frac{1}{2}\mathfrak{t}$.

**Operators**

We now list some interesting gauge invariant operators of the $E[USp(2Q)]$ theory. The fact that they organise themselves into representation of the full $USp(2Q)_y$ symmetry will be a first piece of evidence of the symmetry enhancement.

- Operator $A_x$, transforming in the traceless antisymmetric representation of $USp(2Q)_x$. In our parametrization it has charges $-1$ and $0$ under $U(1)_t$ and $U(1)_c$ respectively and trial R-charge $+2$.

- Operator $A_y$, with the same abelian charges as $A_x$, but which is in the traceless antisymmetric representation of the enhanced $USp(2Q)_y$ symmetry. This is a $2Q \times 2Q$ matrix that can be split in several $2 \times 2$ sub-matrices corresponding to different gauge invariant operators. Those that are placed above the diagonal are constructed starting with one of the diagonal chirals, going along the tail with the bifundamentals and ending on a vertical chiral (see Figure 5). Those below the diagonal are fixed requiring that the matrix is antisymmetric. Finally, the diagonal is filled with the following $Q-1$ operators that are singlets under $USp(2Q)_y$:

$$i\sigma_2 \operatorname{Tr}_{2n} A^{(n)} = \begin{pmatrix} 0 & \operatorname{Tr}_{2n} A^{(n)} \\ -\operatorname{Tr}_{2n} A^{(n)} & 0 \end{pmatrix}, \qquad n = 1, \cdots, Q-1. \tag{17}$$

Notice all that these operators have the same charges under the $U(1)$ symmetries and the same R-charges, so we can collect them in the same matrix $A_y$. As an example, for $Q = 3$ this matrix takes the explicit form

$$A_y = \begin{pmatrix} i\sigma_2 \operatorname{Tr}_2 A^{(1)} & \operatorname{Tr}_2\left(d^{(1)}v^{(1)}\right) & \operatorname{Tr}_2\left[d^{(1)} \operatorname{Tr}_4\left(q_i^{(1,2)\dagger}v^{(2)}\right)\right] \\ -\operatorname{Tr}_2\left(d^{(1)}v^{(1)}\right) & i\sigma_2 \operatorname{Tr}_4 A^{(2)} & \operatorname{Tr}_4\left(d^{(2)}v^{(2)}\right) \\ -\operatorname{Tr}_2\left[d^{(1)} \operatorname{Tr}_4\left(q_i^{(1,2)\dagger}v^{(2)}\right)\right] & -\operatorname{Tr}_4\left(d^{(2)}v^{(2)}\right) & \end{pmatrix}. \tag{18}$$

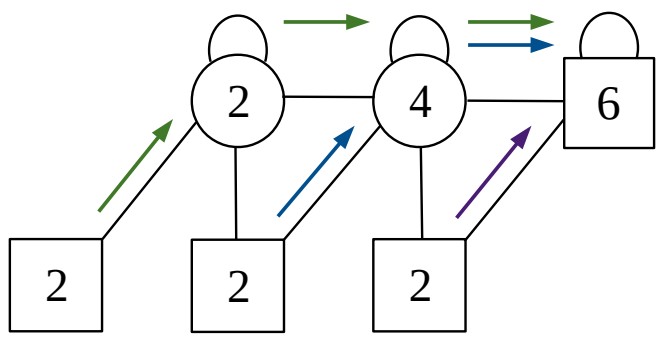

Figure 6: Operators contributing to $\Pi$ in the $Q = 3$ case.

The missing entry in the bottom right corner should be filled in by requiring that the matrix $A_y$ satisfies $\text{Tr}_{2Q} A_y = 0$.

- Operator $\Pi$ in the bifundamental representation of the enhanced $USp(2Q)_x \times USp(2Q)_y$ symmetry. This operator is constructed by collecting all the operators built starting with one diagonal chiral $d^{(n)}$ and going along the tail with all the remaining bifundamentals ending on $q^{(Q-1,Q)}$ (see Figure 6). Each of these operators transform in the fundamental representation of one of the $SU(2)_{y_n}$ symmetries of the saw and in the fundamental representation of $USp(2Q)_x$. Moreover, they all have charges 0 and $+1$ under $U(1)_t$ and $U(1)_c$ respectively and trial R-charge 0. Hence, we can collect them in a vector that becomes an operator transforming in the bifundamental representation of the enhanced $USp(2Q)_x \times USp(2Q)_y$ symmetry. As an example, for $Q = 3$ it takes the explicit form

$$
\Pi = \begin{pmatrix} \text{Tr}_2 \left[ d^{(1)} \text{Tr}_4 \left( q^{(1,2)\dagger} q^{(2,3)\dagger} \right) \right] \\ \text{Tr}_4 \left( d^{(2)} q^{(2,3)\dagger} \right) \\ d^{(3)} \end{pmatrix}. \tag{19}
$$

- There are also other gauge invariant operators that we can construct using the chirals of the saw, which are singlets under $USp(2Q)_x \times USp(2Q)_y$. For example, we have the mesons

$$
\left( \text{Tr}_{2n} d^{(n)} d^{(n)} \right)_{\alpha\beta} = J^{(n)}_{ij} d^{(n)j}{}_\alpha d^{(n)i}{}_\beta. \tag{20}
$$

These are in the antisymmetric representation of $SU(2)_{y_n}$, so they are actually singlets. They also have charges $n - Q$ and $+2$ under $U(1)_t$ and $U(1)_c$ respectively and trial R-charge 0. Moreover, because of the equations of motions of the gauge singlets $b_n$, they are subjected to the following classical relation:

$$
\text{Tr}_2 \left( \text{Tr}_{2n} d^{(n)} d^{(n)} \right) = 0. \tag{21}
$$

To summarize, the main gauge invariant operators and their charges under the global symmetry are

|        | $USp(2Q)_x$ | $USp(2Q)_y$ | $U(1)_t$ | $U(1)_c$ | $U(1)_{R_0}$ |
|--------|-------------|-------------|----------|----------|--------------|
| $A_x$  | antisymm.   | 1           | $-1$     | 0        | 2            |
| $A_y$  | 1           | antisymm.   | $-1$     | 0        | 2            |
| $\Pi$  | $\square$   | $\square$   | 0        | $+1$     | 0            |
| $b_n$  | 1           | 1           | $Q-n$    | $-2$     | 2            |

**The supersymmetric index of $E[USp(2Q)]$**

The supersymmetric index [28–30] of the $E[USp(2Q)]$ theory can be expressed with the following recursive definition (our notations and defintions are collected in Appendix B):

$$
\begin{aligned}
\mathcal{I}_{E[USp(2Q)]}(x_n, y_n, t, c) &= \frac{\prod_{n=1}^{Q} \Gamma_e\left(c\, y_Q^{\pm 1} x_n^{\pm 1}\right)}{\Gamma_e\left(c^2\right) \Gamma_e(t)^{Q-1} \prod_{n<m}^{Q} \Gamma_e\left(t\, x_n^{\pm 1} x_m^{\pm 1}\right)} \times \\
&\quad \times \oint \mathrm{d}\vec{z}_{Q-1} \frac{\mathcal{I}_{E[USp(2(Q-1))]}(z_1, \cdots, z_{Q-1}, y_1, \cdots, y_{Q-1}, t, t^{-1/2}c)}{\Gamma_e(t) \prod_{i<j}^{Q-1} \Gamma_e\left(z_i^{\pm 1} z_j^{\pm 1}\right) \prod_{i=1}^{Q-1} \Gamma_e\left(z_i^{\pm 2}\right)} \times \\
&\quad \times \frac{\prod_{i=1}^{Q-1} \prod_{n=1}^{Q} \Gamma_e\left(t^{1/2} z_i^{\pm 1} x_n^{\pm 1}\right)}{\prod_{i=1}^{Q-1} \Gamma_e\left(t^{1/2} c\, y_Q^{\pm 1} z_i^{\pm 1}\right)},
\end{aligned}
\tag{22}
$$

where we defined the integration measure as

$$
\mathrm{d}\vec{z}_n = \frac{[(p;p)(q;q)]^n}{2^n n!} \prod_{i=1}^{n} \frac{\mathrm{d}z_i}{2\pi i\, z_i}.
\tag{23}
$$

To define the index we use the assignment of R-charges as depicted in Figure 4. If one wishes to use the superconformal assignment of R-charges then the parameters should be redefined as,

$$
c \to c\, (pq)^{\mathfrak{c}/2}, \qquad t \to t\, (pq)^{\mathfrak{t}/2},
\tag{24}
$$

where $\mathfrak{c}$ and $\mathfrak{t}$ are the mixing coefficients appearing in eq. (10).

This expression coincides with the interpolation kernel $\mathcal{K}_c(x, y)$ recently introduced in [32] as an analytic continuation of the elliptic interpolation functions.[7]

## 3.2 IR dualities

The kernel $\mathcal{K}_c(x, y)$ satisfies various remarkable properties leading to integral identities which below we reinterpret as highly non-trivial dualities for the $E[USp(2Q)]$ theory.

**Duality I: self-duality**

The first duality involving the $E[USp(2Q)]$ theory is actually a self-duality. Rains has proven in [32] that the supersymmetric index of $E[USp(2Q)]$ first satisfies the property that the fugacities of the $SU(2)_y^Q$ symmetry are actually forming always characters of $USp(2Q)_y$ symmetry suggesting the enhancement of symmetry. Second, the index is invariant under exchanging the $USp(2Q)_x$ and the $USp(2Q)_y$ fugacities suggesting a self-duality of the theory. This duality

---

[7]Notice that the antisymmetric of the $USp(2Q)_x$ flavor symmetry is traceless. However at each step of the iteration we complete the antisymmetric of the gauged $USp(2(Q-1))$ symmetry with a singlet corresponding to its trace. This differs slightly from the recursive definition of $\mathcal{K}_c(x, y)$ in [32], where each $USp(2n)$ node, including the last ungauged node, has a full antisymmetric.

acts by exchanging the operators charged under the $USp(2Q)_x$ and the enhanced $USp(2Q)_y$ symmetries

$$\begin{aligned} A_x &\leftrightarrow A_y, \\ \Pi &\leftrightarrow \Pi^\dagger. \end{aligned} \tag{25}$$

At the level of the index we have the following identity:

$$\mathcal{I}_{E[USp(2Q)]}(x_n, y_n, t, c) = \mathcal{I}_{E[USp(2Q)]}(y_n, x_n, t, c), \tag{26}$$

which has been proven in Theorem 3.1 of [32]. Again, from this duality we can clearly see the symmetry enhancement, since the supersymmetric index is invariant under the exchange of the fugacities of $USp(2Q)_x$ and $\prod_{n=1}^{Q} SU(2)_{y_n}$.

**Duality II: flip-flip duality**

The $E[USp(2Q)]$ theory also enjoys another duality that maps it to another $E[USp(2Q)]$ theory with two extra sets of singlets $F_x$ and $F_y$ in the traceless antisymmetric representation of $USp(2Q)_x$ and $USp(2Q)_y$ respectively, which flip the gauge invariant operators $A_x$ and $A_y$

$$\delta\mathcal{W} = \mathrm{Tr}_{2Q}\left(F_x A_x + F_y A_y\right). \tag{27}$$

Notice that since $A_x$ is a fundamental field in the UV description, $A_x$ and $F_x$ are massive and can be integrated out. We end up with $E[USp(2Q)]$ with a gauge singlet $F_y$ in the antisymmetric representation of $USp(2Q)_y$ rather then $A_x$ in the antisymmetric of $USp(2Q)_x$ and superpotential

$$\begin{aligned} \mathcal{W} &= \mathrm{Tr}_2\left(A^{(1)}\mathbb{M}_R^{(1)}\right) + \sum_{n=2}^{Q-1} \mathrm{Tr}_{2n}\left[A^{(n)}\left(\mathbb{M}_R^{(n)} - \mathbb{M}_L^{(n)}\right)\right] - \mathrm{Tr}_{2Q}\left(F_y A_y\right) + \\ &+ \sum_{n=1}^{Q-1} \mathrm{Tr}_2\, \mathrm{Tr}_{2n}\, \mathrm{Tr}_{2(n+1)}\left(v^{(n)} q^{(n,n+1)} d^{(n+1)}\right) + \sum_{n=1}^{N} b_n\, \mathrm{Tr}_2\, \mathrm{Tr}_{2n}\left(d^{(n)} d^{(n)}\right). \end{aligned} \tag{28}$$

The duality acts leaving unchanged the two $USp(2Q)$ symmetries as well as $U(1)_c$, while inverting the fugacities of the $U(1)_t$ symmetry with its mixing coefficient $t$ changing to $2 - t$. Accordingly, operators are mapped across the duality as follows:

$$\begin{aligned} A_x &\leftrightarrow \mathbb{M}_L^{(N)}, \\ A_y &\leftrightarrow F_y, \\ \Pi &\leftrightarrow \Pi. \end{aligned} \tag{29}$$

This duality is reminiscent of the flip-flip duality of the $FT[SU(Q)]$ theory discussed in [26]. Indeed, as we will see in section 5, $E[USp(2Q)]$ reduces to $FT[SU(Q)]$ upon 3d compactification and a consecutive real mass flow. It should be possible to derive this duality by sequentially applying Intriligator-Pouliot duality [54] starting from the first node.

At the level of the supersymmetric index, the flip-flip duality is encoded in the following integral identity:

$$\begin{aligned} \mathcal{I}_{E[USp(2Q)]}(x_n, y_n, pq/t, c) &= \Gamma_e(t)^{2(Q-1)} \prod_{n<m}^{Q} \Gamma_e\left(t\, x_n^{\pm 1} x_m^{\pm 1}\right) \Gamma_e\left(t\, y_n^{\pm 1} y_m^{\pm 1}\right) \times \\ &\times \mathcal{I}_{E[USp(2Q)]}(x_n, y_n, t, c), \end{aligned} \tag{30}$$

which is proven in Proposition 3.5 of [32]. This duality will play an important role in the relation to E-string compactifications in the next section. In particular we will see that the $U(1)_t$ symmetry is related to the Cartan of the $SU(2)_L$ symmetry factor of rank $Q$ E-string and the duality is the statement that the $U(1)_t$ symmetry enhances to $SU(2)_L$ in that context.

## 3.3 Interesting properties under RG flows

In addition to dualities the $E[USp(2Q)]$ theory enjoys interesting properties under RG flows triggered by turning on vacuum expectation values to various operators.

### $E[USp(2Q)]$ to $E[USp(2(Q-1))]$

The $E[USp(2Q)]$ quiver theory reduces to a smaller quiver tail when a suitable deformation that breaks $USp(2Q)_x \times USp(2Q)_y \to USp(2(Q-1))_x \times USp(2(Q-1))_y$ is taken. More precisely, the deformation in question corresponds to a minimal VEV for the operator $\Pi$, i.e. $\langle \Pi_{2Q,2Q} \rangle \neq 0$. This can be achieved by introducing an additional singlet field that flips this operator and turning on such singlet linearly in the superpotential. The equations of motion of the singlet then imply that the operator acquired a non-vanishing VEV.

At the level of the supersymmetric index, this deformation implies the constraint $x_Q = c\,y_Q$, for which we have (see Lemma 3.1 of [32])

$$\lim_{x_Q \to c\,y_Q} \mathcal{I}_{E[USp(2Q)]}(x_n, y_n, t, c) \frac{\Gamma_e(t)\,\Gamma_e(c^2)}{\Gamma_e\left(c\,x_Q^{\pm 1} y_Q^{\pm 1}\right)} =$$

$$= \prod_{n=1}^{Q} \frac{\Gamma_e\left(c\,y_Q x_n^{\pm 1}\right)\Gamma_e\left(y_Q^{-1} y_n^{\pm 1}\right)}{\Gamma_e\left(t\,c\,y_Q x_n^{\pm 1}\right)\Gamma_e\left(t\,y_Q^{-1} y_n^{\pm 1}\right)} \mathcal{I}_{E[USp(2(Q-1))]}(x_n, y_n, t, c). \tag{31}$$

### $E[USp(2Q)]$ to WZ

If we add a linear term $\delta\mathcal{W} = b_{Q-1}$ to the superpotential, the second last diagonal flavor $d^{(Q-1)}$ takes a VEV and the last gauge node is partially Higgsed. The $E[USp(2Q)]$ theory reduces to a bifundamental of $USp(2Q)_y \times USp(2Q)_x$ with the two flavor nodes flipped by traceless antisymmetric representations.

It is easy how this work in the $Q = 2$ case, where the index is given by:

$$\mathcal{I}_{E[USp(4)]}(x_1, x_2, y_1, y_2, t, c) =$$

$$= \frac{\prod_{n=1}^{2} \Gamma_e\left(c\,y_2^{\pm 1} x_n^{\pm 1}\right)}{\Gamma_e(c^2)\,\Gamma_e(t^{-1}c^2)\,\Gamma_e(t)^2\,\Gamma_e\left(t\,x_1^{\pm 1} x_2^{\pm 1}\right)} \times$$

$$\times \oint dz_1 \frac{\Gamma_e\left(t^{-1/2} c\,z^{\pm 1} y_1^{\pm 1}\right) \prod_{n=1}^{2} \Gamma_e\left(t^{1/2} z^{\pm 1} x_n^{\pm 1}\right)}{\Gamma_e(z^{\pm 2})\,\Gamma_e(t^{1/2} c\,z^{\pm 1} y_2^{\pm 1})}. \tag{32}$$

$$\tag{33}$$

The condition of $b_1$ entering the superpotential corresponds to $c \to \sqrt{t}$. In this limit the poles of the integrand at $z = t^{1/2} c^{-1} y_1^{\pm}$ and $z = t^{-1/2} c\,y_1^{\pm}$ pinch the integration contour in two points and we can evaluate the index by taking the residue at these two points as in [55]. Both poles give the same contribution to the index and we get:

$$\lim_{c \to \sqrt{t}} \mathcal{I}_{E[USp(4)]}(x_n, y_n, t, c) = \frac{\prod_{n,m=1}^{2} \Gamma_e(\sqrt{t} x_n^{\pm} y_m^{\pm})}{\Gamma_e(t)^2\,\Gamma_e(t x_1^{\pm} x_2^{\pm})\Gamma_2(t y_1^{\pm} y_2^{\pm})}. \tag{34}$$

At higher rank, the condition of $b_{Q-1}$ entering the superpotential still corresponds to $c \to \sqrt{t}$ and the reduction of the index follows by Proposition 3.5 in [32]:

$$\lim_{c \to \sqrt{t}} \mathcal{I}_{E[USp(2Q)]}(x_n, y_n, t, c) = \frac{\prod_{n,m=1}^{Q} \Gamma_e(\sqrt{t} x_n^{\pm} y_m^{\pm})}{\Gamma_e(t)^{2(Q-1)} \prod_{n<m}^{Q} \Gamma_e(t x_n^{\pm} x_m^{\pm})\Gamma_e(t y_n^{\pm} y_m^{\pm})}. \tag{35}$$

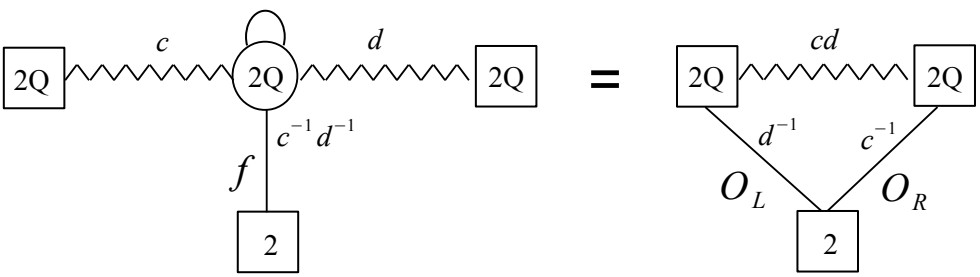

Figure 7: Schematic representation of the braid relation.

## 3.4 Braid relation: generalized Seiberg duality

The $E[USp(2Q)]$ theory has another interesting property. If we glue two $E[USp(2Q)]$ blocks by gauging one of the two $USp(2Q)$ symmetries of each tail together with an extra flavor $f$ charged under this gauge symmetry we re-obtain $E[USp(2Q)]$ plus two extra sets of singlets $O_L$ and $O_R$, as depicted in Figure 7. Notice that for $Q = 1$ the braid relation reduces to Seiberg duality for $SU(2)$ with 3 flavors, which is dual to a WZ model.

At the level of the index this duality is encoded in the following braid relation given in Proposition 2.12 of [32]:

$$\oint d\vec{z}_Q\, \mathcal{I}_{E[USp(2Q)]}(x_n, z_n, t, c)\mathcal{I}_{E[USp(2Q)]}(z_n, y_n, t, d) \frac{\Gamma_e(t)^{Q-1} \prod_{n<m}^Q \Gamma_e\left(t\, z_n^{\pm 1} z_m^{\pm 1}\right)}{\prod_{n=1}^Q \Gamma_e\left(z_n^{\pm 2}\right) \prod_{n<m}^Q \Gamma_e\left(z_n^{\pm 1} z_m^{\pm 1}\right)} \times$$

$$\times \prod_{n=1}^Q \Gamma_e\left(u_0 z_n^{\pm 1}, u_1 z_n^{\pm 1}\right) = \tag{36}$$

$$= \prod_{n=1}^Q \Gamma_e\left(c\, u_0 x_n^{\pm 1}, c\, u_1 x_n^{\pm 1}, d\, u_0 y_n^{\pm 1}, d\, u_1 y_n^{\pm 1}\right) \mathcal{I}_{E[USp(2Q)]}(x_n, y_n, t, cd),$$

$$\tag{37}$$

which holds if the following balancing condition is satisfied:

$$u_0 u_1 = \frac{pq}{c^2 d^2}. \tag{38}$$

Let's discuss in more details the superpotential of the dual theories and their gauge invariant operators. We first consider the l.h.s. of the duality where we glue two $E[USp(2Q)]$ blocks. We name the fields as in Figure 8.

In this case the superpotential is the one of the two $E[USp(2Q)]$ tails

$$\mathcal{W} = \mathcal{W}^L_{E[USp(2Q)]} + \mathcal{W}^R_{E[USp(2Q)]}, \tag{39}$$

where in the middle $USp(2Q)$ gauge node we have only one traceless antisymmetric that couples both to $q_L^{(Q-1,Q)}$ and $q_L^{(Q-1,Q)}$, so in (39) we have to identify $A_L^{(Q)} = A_R^{(Q)}$.

The balancing condition here follows by requiring $U(1)_R$ to be non-anomalous at the central $USp(2Q)$ node:[8]

$$Q + 1 + (Q-1)(1-\mathfrak{t}) + 2(Q-1)(\tfrac{\mathfrak{t}}{2} - 1) + (\mathfrak{d} - 1) + (\mathfrak{c} - 1) + \tfrac{1}{2}(\mathfrak{u}_0 + \mathfrak{u}_1 - 1) = 0. \tag{40}$$

---

[8]The relation between $t, c, d, u_0, u_1$ and $\mathfrak{t}, \mathfrak{c}, \mathfrak{d}, \mathfrak{u}_0, \mathfrak{u}_1$ is as in eq. (24).

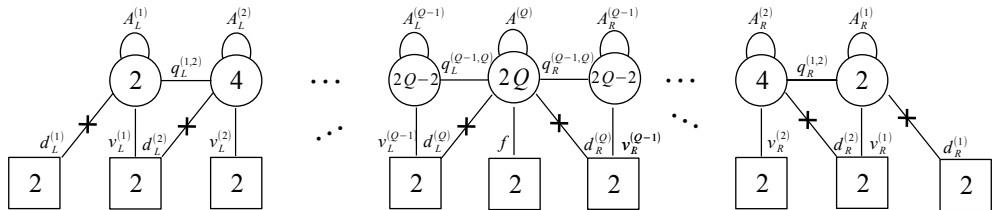

Figure 8: Fields appearing in the lhs of the braid relation.

If we rescale the fugacities for the two chirals $f_1$, $f_2$ as

$$u_0 \to u_0 \frac{\sqrt{pq}}{cd}, \qquad u_1 \to u_1 \frac{\sqrt{pq}}{cd}, \tag{41}$$

we can see that the balancing condition (38) becomes the standard tracelessness condition $u_0 u_1 = 1$ for the $SU(2)$ symmetry rotating them. Hence, the (enhanced) global symmetry is

$$USp(2Q)_x \times USp(2Q)_y \times U(1)_t \times U(1)_c \times U(1)_d \times SU(2)_u. \tag{42}$$

The gauge invariant operators are constructed starting from those of the two $E[USp(2Q)]$ tails:

- operators $A_x^L$, $A_y^R$ constructed as in (18) using the diagonal, vertical and bifundamental chirals of the left and right $E[USp(2Q)]$ blocks respectively;

- operator $\Xi$ constructed starting from one of the diagonals of the left $E[USp(2Q)]$ and terminating on a diagonal of the right $E[USp(2Q)]$ including bifundamentals. For $Q = 2$ this is given by:

$$\Xi = \begin{pmatrix} \mathrm{Tr}_{2,L}\,\mathrm{Tr}_{2,R}\left[d_L^{(1)}\,\mathrm{Tr}_4\left(q_L^{(1,2)\dagger}q_R^{(1,2)}\right)d_R^{(1)\dagger}\right] & \mathrm{Tr}_{2,L}\left[d_L^{(1)}\,\mathrm{Tr}_4\left(q_L^{(1,2)\dagger}d_R^{(2)\dagger}\right)\right] \\ \mathrm{Tr}_{2,R}\left[\mathrm{Tr}_4\left(d_L^{(2)}q_R^{(1,2)}\right)d_R^{(1)\dagger}\right] & \mathrm{Tr}_4\left[d_L^{(2)}d_R^{(2)\dagger}\right] \end{pmatrix}; \tag{43}$$

- operators $\Omega^L$, $\Omega^R$ constructed by joining the operators $\Pi^L$ and $\Pi^R$ in the bifundamental representation of $USp(2Q)_x \times USp(2Q)_z$ and $USp(2Q)_y \times USp(2Q)_z$ respectively with the two fundamental chirals $f_i$

$$\Omega^L = \mathrm{Tr}_{2Q}\left(\Pi^L f\right), \qquad \Omega^R = \mathrm{Tr}_{2Q}\left(\Pi^R f\right); \tag{44}$$

- long mesons constructed with the bifundamentals and the chirals $f_i$

$$\Theta_n = \mathrm{Tr}_2\,\mathrm{Tr}_{2n}\left[\mathrm{Tr}_{2Q}\left(f^\dagger \prod_{a=n}^{Q-1} q_{L/R}^{(a,a+1)\dagger}\right)\mathrm{Tr}_{2Q}\left(f \prod_{a=n}^{Q-1} q_{L/R}^{(a,a+1)}\right)\right], \tag{45}$$

where $L/R$ means that we can construct two sets of operators of this form with the bifunamentals of the left or the right tail respectively, but they actually coincide in pairs in the chiral ring;

- flipping fields $b_n^L$, $b_n^R$ of the diagonal mesons on the left and right $E[USp(2Q)]$ tails.

|  | $USp(2Q)_x$ | $USp(2Q)_y$ | $U(1)_t$ | $U(1)_c$ | $U(1)_d$ | $SU(2)_u$ | $U(1)_{R_0}$ |
|---|---|---|---|---|---|---|---|
| $A_x^L$ | antisymm. | 1 | $-1$ | 0 | 0 | 1 | 2 |
| $A_y^R$ | 1 | antisymm. | $-1$ | 0 | 0 | 1 | 2 |
| $\Xi$ | □ | □ | 0 | 1 | 1 | 1 | 0 |
| $\Omega^L$ | □ | 1 | 0 | $-1$ | 0 | □ | 1 |
| $\Omega^R$ | 1 | □ | 0 | 0 | $-1$ | □ | 1 |
| $b_n^L$ | 1 | 1 | $Q-n$ | $-2$ | 0 | 1 | 2 |
| $b_n^R$ | 1 | 1 | $Q-n$ | 0 | $-2$ | 1 | 2 |
| $\Theta_n$ | 1 | 1 | $Q-n$ | $-2$ | $-2$ | 1 | 2 |

The charges of these operators under the global symmetries are summarized in the following table:

On the r.h.s. we have $E[USp(2Q)]$ with two sets of chiral singlets $O_L$ and $O_R$ in the bifundamental representation of the global $USp(2Q)_x \times SU(2)_u$ and $SU(2)_u \times USp(2Q)_y$ symmetries respectively, which interact with the $E[USp(2Q)]$ block through the superpotential

$$\mathcal{W} = \mathcal{W}_{E[USp(2Q)]} + \text{Tr}_2 \, \text{Tr}_{2Q_x} \, \text{Tr}_{2Q_y} \, O^L \Pi O^R \,. \tag{46}$$

Because of this superpotential, the global symmetry of the theory precisely matches with (42).

The gauge invariant operators are the same of $E[USp(2Q)]$ with the addition of the two sets of singlets $O_L$ and $O_R$. Moreover, we can construct some long mesons of the form

$$\Theta_n^R = \text{Tr}_2 \, \text{Tr}_{2n} \left[ \text{Tr}_{2Q} \left( O_R^\dagger \prod_{a=n}^{Q-1} q^{(a,a+1)\dagger} \right) \text{Tr}_{2Q} \left( O_R \prod_{a=n}^{Q-1} q^{(a,a+1)} \right) \right] \tag{47}$$

and similar ones $\Theta_n^L$ involving $O_L$ which have not a simple expression in terms of fundamental fields but whose existence is guaranteed by the self-duality of the $E[USp(2Q)]$ block. Their charges under the global symmetries are

|  | $USp(2Q)_x$ | $USp(2Q)_y$ | $U(1)_t$ | $U(1)_c$ | $U(1)_d$ | $SU(2)_u$ | $U(1)_{R_0}$ |
|---|---|---|---|---|---|---|---|
| $A_x$ | antisymm. | 1 | $-1$ | 0 | 0 | 1 | 2 |
| $A_y$ | 1 | antisymm. | $-1$ | 0 | 0 | 1 | 2 |
| $\Pi$ | □ | □ | 0 | 1 | 1 | 1 | 0 |
| $b_n$ | 1 | 1 | $Q-n$ | $-2$ | $-2$ | 1 | 2 |
| $O^L$ | □ | 1 | 0 | 0 | $-1$ | □ | 1 |
| $O^R$ | 1 | □ | 0 | $-1$ | 0 | □ | 1 |
| $\Theta_n^R$ | 1 | 1 | $Q-n$ | $-2$ | 0 | 1 | 2 |
| $\Theta_n^L$ | 1 | 1 | $Q-n$ | 0 | $-2$ | 1 | 2 |

The operator map across the duality is then

$$\begin{aligned}
A_x^L &\leftrightarrow A_x \,, \\
A_y^R &\leftrightarrow A_y \,, \\
\Xi &\leftrightarrow \Pi \,, \\
\Omega^L &\leftrightarrow O^L \,, \\
\Omega^R &\leftrightarrow O^R \,, \\
\Theta_n &\leftrightarrow b_n \,, \\
b_n^R &\leftrightarrow \Theta_n^L \,, \\
b_n^L &\leftrightarrow \Theta_n^R \,.
\end{aligned} \tag{48}$$

# 4   Rank $Q$ E-string on tubes and tori

We are interested in constructing four dimensional theories which flow to theories one would obtain by compactifying the rank $Q$ E-string theory either on a two punctured sphere or a torus with some flux in the abelian subgroups of the $E_8$ symmetry factor of the six dimensional theory. We have already discussed some general expectations from such theories, such as symmetries and anomalies, in section 2.1. We will show in this section that such four dimensional theories can be constructed using $E[USp(2Q)]$ as an essential basic building block.

Theories corresponding to tubes and tori can be constructed by gluing together theories corresponding to compactifications on tubes with some minimal value of flux, which we will refer to as *the tube* model. The precise meaning of this statement will be discussed next. We note that the following discussion is an abstraction of rules that were observed to work in various examples, notably the rank 1 E-string [1], and some of its generalizations [16]. While there are arguments in favor of this picture, it is ultimately motivated mostly by observation.

## 4.1   The basic tube and gluing

Let us start from a geometric definition of the tube model. The tube model we discuss here is a four dimensional theory corresponding to the compactification of the 6$d$ E-string theory on a two punctured sphere with some particular value of flux for the $E_8$ symmetry. The punctures have $USp(2Q)$ symmetry associated to them and they come in different types. These come about since the punctures are expected to break the $E_8$ global symmetry to $SO(16)$, and we have some freedom in how the $SO(16)$ is embedded inside the $E_8$. Specifically, we are free to act with any inner automorphism of $E_8$, which are just the Weyl transformations, to potentially get different embeddings. Likewise the flux, being a vector in the root lattice of $E_8$, is also affected by Weyl transformations. Thus, given a tube we can generate an equivalent tube by acting with an $E_8$ Weyl transformation. Tubes differing in this way are ultimately the same tube, but the gluing of two tubes is affected if these differ by a relative Weyl group action. When we glue two punctures together the associated flux for the combined tube is the sum with appropriate signs, as we shall see, of their fluxes.

Now we need to define the tube and gluing at the level of the physical theories. In a tube theory each puncture comes equipped with an octet of fundamental operators of $USp(2Q)$, $M_i$, which we will refer to by an abuse of notation as moment maps. These moment map operators are charged under the $U(1)$ symmetries comprising the Cartan generators of the $E_8$. Different types of punctures have moment map operators charged differently under these symmetries. Again, the difference of the charges can be associated to an action of the Weyl group of $E_8$. Consider fixing a specific $SO(16)$ subgroup of $E_8$, as we have done when we chose an $SO(16)$ flux basis. Then for simplicity, when we glue two punctures together we can first limit ourselves to only gluing punctures of the same type up to the action of the Weyl group of the chosen $SO(16)$ subgroup of $E_8$ (a more general gluing will be discussed in section 4.4). The Weyl group of $SO(16)$ is comprised of permutations of eight elements and of flips of any even number of them. In particular it means that the moment map operators $M_i$ and $M_i'$ of the two punctures we are gluing have same charges under the Cartan symmetries of $E_8$ up to permutations of the indices and flips of signs for even number of components. Let us denote the permutation by $\sigma$ and the set of indices with flipped signs by $\mathfrak{F}$. The punctures also have operators $A$ in the antisymmetric traceless representation of $USp(2Q)$. We glue punctures by gauging a diagonal combination of the two $USp(2Q)$ symmetries and by introducing a chiral field, $\hat{A}$, in the traceless antisymmetric representation of $USp(2Q)$ and a set of fundamental

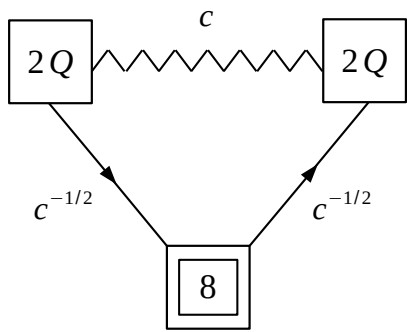

Figure 9: The basic tube with $E_7$ flux $\mathcal{F} = (\frac{1}{2}, \frac{1}{2}, \frac{1}{2}, \frac{1}{2}, \frac{1}{2}, \frac{1}{2}, \frac{1}{2}, \frac{1}{2})$. The squares denote $USp$ groups and double squares $SU$ groups.

fields, $\Phi_i$ for $i \notin \mathfrak{F}$, which couple to the moment map operators through a superpotential,

$$\mathcal{W} = \sum_{j \notin \mathfrak{F}} (M_j - M'_{\sigma(j)}) \Phi^j + \sum_{j \in \mathfrak{F}} M_j M'_{\sigma(j)} + (A - A')\hat{A}. \tag{49}$$

The first type of superpotential terms was referred to as $\Phi$ gluing and the second ones as $S$ gluing in [1, 17]. The third term only appears for higher rank E-string as for rank one we do not have traceless antisymmetric representations. Physically the restriction to only having even number of flipped charges is related to Witten global anomaly obstruction [56]. We will be gauging the $USp(2Q)$ symmetry and the absence of the Witten anomaly implies that the number of chiral fields in the fundamental representation here is even. Finally if the fluxes of the theories we are gluing are $\mathcal{F}$ and $\mathcal{F}'$, the flux of the combined theory, $\mathcal{F}^{glued}$, will be given by

$$i \notin \mathfrak{F} : \mathcal{F}_i^{glued} = \mathcal{F}_i + \mathcal{F}'_{\sigma(i)}, \qquad i \in \mathfrak{F} : \mathcal{F}_i^{glued} = \mathcal{F}_i - \mathcal{F}'_{\sigma(i)}. \tag{50}$$

Next we need to define at least one tube model from which we can build other tubes and torus theories. The simplest tube, depicted in Figure 9, is constructed by coupling the $E[USp(2Q)]$ block to two octets $M, M'$ with superpotential

$$\mathcal{W} = \sum_{a=1}^{8} M^a \Pi M'_a. \tag{51}$$

This tube model is associated to a flux breaking $E_8 \rightarrow U(1)_c \times E_7$ which in the $SO(2)^8 \subset SO(16)$ basis corresponds to the vector,

$$\mathcal{F} = \left( \frac{1}{2}, \frac{1}{2}, \frac{1}{2}, \frac{1}{2}, \frac{1}{2}, \frac{1}{2}, \frac{1}{2}, \frac{1}{2} \right). \tag{52}$$

The basic tube theory has global symmetry $USp(2Q) \times USp(2Q) \times U(1)_t \times U(1)_c \times SU(8) \times U(1)_A$. The $U(1)_t$ symmetry is hidden inside the block and when we glue tubes into tori it will enhance to the $SU(2)_L$ symmetry of the E-string. The $U(1)_A$ symmetry, when we glue tubes into tori, always disappears because of anomalies and superpotential constraints. For this reason we will omit it from our discussion as it appears to be accidental from the six dimensional point of view. We use the $U(1)_c$ fugacity to define charges of the moment maps on the right as $a_i = c^{-1/2} u_i$, where $u_i$ are $SU(8)$ fugacities satisfying $\prod_{i=1}^{8} u_i = 1$. On the left the fugacities

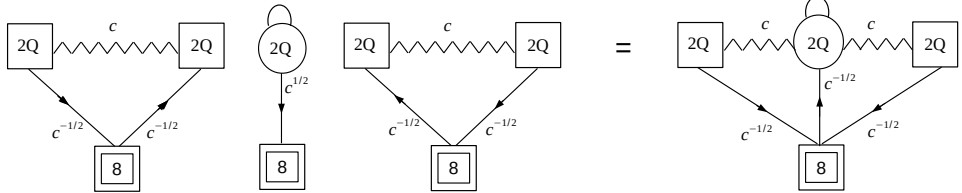

Figure 10: Gluing two basic tubes together with a trivial element of the Weyl group we obtain tube with flux $(1, 1, 1, 1, 1, 1, 1, 1)$.

are $a_i' = c^{-1/2}u_i^{-1}$. The map between $a_i$ to $a_i'$ consists of charge conjugation for the $SU(8)$, but without acting on $U(1)_c$. This is not a Weyl group element of $SO(16)$, which contains charge conjugation for both groups, but not for each one separately. However, as we explain in appendix A, it is an element of the $E_7 \subset E_8$ Weyl symmetry group.

Let us illustrate how we glue two such tubes together with concrete examples. We can glue two basic tubes with a trivial identification of the moment maps as depicted in Figure 10. If we assign fugacities $a_i$ to moment maps of one glued puncture and $b_i$ to the other glued puncture, we identify the charges with a trivial action of the $SO(16)$ Weyl symmetry group:

$$i = 1\ldots 8 : \ a_i = b_i. \tag{53}$$

The two basic tubes then are glued with the superpotential

$$\mathcal{W} = \sum_{j=1}^{8}(M_j - M'_{\sigma(j)})\Phi^j + (A - A')\hat{A}. \tag{54}$$

Integrating out the massive fields, the fields $M_j$ and $M'_{\sigma(j)}$ are identified and we get the quiver on the right of Figure 10. The flux of the combined model is obtained summing the fluxes $\mathcal{F}$ and $\mathcal{F}'$ of the two glued theories. Since in this case there are no flips of fugacities the tube model which we obtain has flux

$$(\mathcal{F}_1 + \mathcal{F}'_1, \mathcal{F}_2 + \mathcal{F}'_2, \mathcal{F}_3 + \mathcal{F}'_3, \mathcal{F}_4 + \mathcal{F}'_4, \mathcal{F}_5 + \mathcal{F}'_5, \mathcal{F}_6 + \mathcal{F}'_6, \mathcal{F}_7 + \mathcal{F}'_7, \mathcal{F}_8 + \mathcal{F}'_8) = \tag{55}$$
$$= (1, 1, 1, 1, 1, 1, 1, 1),$$

which corresponds to a unit of flux $z = 1$ for the $U(1)$ whose commutant in $E_8$ is $E_7$.

We can also glue two basic tubes with a non-trivial identification of the moment maps. Denoting again as $a_i$ and $b_i$ the fugacities of the punctures we are gluing, we identify the charges with an action of the $SO(16)$ Weyl symmetry group,

$$i = 1\ldots 4 : \ a_i = b_i, \tag{56}$$
$$i = 5\ldots 8 : \ a_i = 1/b_i.$$

We decompose the $SU(8) \times U(1)_c$ fugacities of our basic tube into $SU(4) \times SU(4) \times U(1)_s \times U(1)_c$ fugacities taking for the first moment map fugacities $a_i = c^{-\frac{1}{2}}s^{-\frac{1}{2}}v_i$ for $i = 1\ldots 4$ and $a_i = c^{-\frac{1}{2}}s^{\frac{1}{2}}w_i$ for $i = 5\ldots 8$, with $\prod_{i=1}^{4}v_i = \prod_{i=5}^{8}w_i = 1$. Analogously for the second moment map we take $b_i = c'^{-\frac{1}{2}}s'^{-\frac{1}{2}}v_i'$ for $i = 1\ldots 4$ and $b_i = c'^{-\frac{1}{2}}s'^{\frac{1}{2}}w_i'$ for $i = 5\ldots 8$. The identification above then sets $c = s'$, $s = c'$, $v_i = v_i'$, and $w_i = 1/w_i'$. The gluing of the two tubes is depicted in Figure 11. The two basic tubes are now glued with the superpotential

$$\mathcal{W} = \sum_{j=1}^{4}(M_j - M'_{\sigma(j)})\Phi^j + \sum_{j=5}^{8}M_j M'_{\sigma(j)} + (A - A')\hat{A}. \tag{57}$$

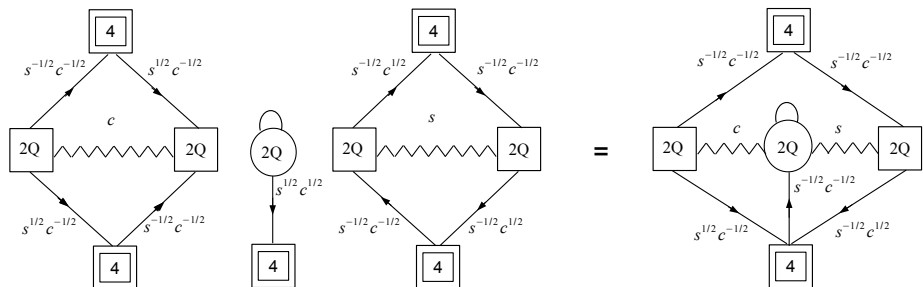

Figure 11: Example of gluing two basic tubes together with a non-trivial element of the Weyl group of $SO(16)$. The resulting tube will have flux $(1,1,1,1,0,0,0,0)$.

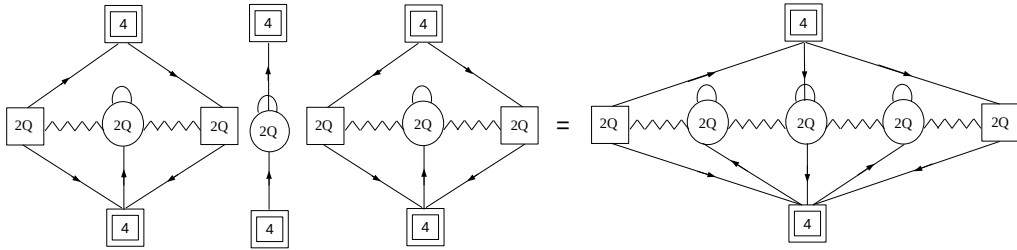

Figure 12: Gluing two $(1,1,1,1,0,0,0,0)$ tubes together with a trivial element of the Weyl group of $SO(16)$ we obtain a tube with flux $(2,2,2,2,0,0,0,0)$.

Now half of the fugacities are flipped and consequently the tube model we obtain has flux

$$
\begin{aligned}
&(\mathcal{F}_1 + \mathcal{F}'_1, \mathcal{F}_2 + \mathcal{F}'_2, \mathcal{F}_3 + \mathcal{F}'_3, \mathcal{F}_4 + \mathcal{F}'_4, \mathcal{F}_5 - \mathcal{F}'_5, \mathcal{F}_6 - \mathcal{F}'_6, \mathcal{F}_7 - \mathcal{F}'_7, \mathcal{F}_8 - \mathcal{F}'_8) \\
&= (1,1,1,1,0,0,0,0),
\end{aligned}
\tag{58}
$$

which corresponds to half a unit of flux $z = \frac{1}{2}$ for the $U(1)$ whose commutant in $E_8$ is $SO(14)$.

We can further glue these tubes. For example, by gluing two $(1,1,1,1,0,0,0,0)$ tubes with a trivial action of the $SO(16)$ Weyl symmetry group, adding eight $\Phi_i$ fundamentals, as shown in Figure 12, we obtain a tube with flux $(2,2,2,2,0,0,0,0)$ which corresponds to a unit of flux $z = 1$ for the $U(1)$ whose commutant in $E_8$ is $SO(14)$.

Using these simple definitions we will now construct a large set of models with interesting properties. Before doing this we will verify that the 't Hooft anomalies under all the symmetries of the conjectured tube theory match the six dimensional predictions (1), (3), (4), and (5).

Let us also note here that the RG flow between $E[USp(2Q)]$ to $E[USp(2(Q-1))]$ of $E[USp(2Q)]$ that we have discussed in section 3.3 has a $6d$ meaning. This flow corresponds to separating one $M5$ brane from the rest and flowing to a lower rank E-string theory. Note that such a flow keeps the six dimensional symmetry $E_8 \times SU(2)_L$ intact. However as the symmetries corresponding to the punctures in the $E[USp(2Q)]$ and $E[USp(2(Q-1))]$ are different, the VEV breaks $USp(2Q)$ down to $USp(2(Q-1))$. We also note that the flow to WZ model discussed in 3.3 was considered in the context of rank $Q$ E-string compactification in Appendix B of [1] and corresponds to some relevant deformation of the theories obtained in the compactifications.

## 4.2 Anomalies of the basic tube

Let us compute various anomalies of the tube theory. We have defined the basic model using a certain $R$ symmetry and definition of $U(1)_c$ and $U(1)_t$ using which the charges of various fields take the simplest form. Also these are the definitions used by Rains in [32]. However, to compare with $6d$ computation we need to perform slight redefinitions. In general, as we mentioned before, different choices of R-symmetry are related by admixture of abelian symmetries

$$R = R_0 + \mathfrak{c} q_c + \mathfrak{t} q_t.$$

Here we will use the six dimensional R-symmetry, which we will denote by $\hat{R}$, which corresponds to taking $\mathfrak{c} = 0$ and $\mathfrak{t} = 1$ so $\hat{R} = R_0 + q_t$. Using this R-symmetry we find that the linear anomalies are

$$\text{Tr}\, U(1)_{\hat{R}} = -Q(1 + 2Q), \qquad \text{Tr}\, U(1)_t = 1 + Q - 2Q^2, \qquad \text{Tr}\, U(1)_c = -14Q. \tag{59}$$

Next consider anomalies with puncture symmetries

$$\text{Tr}\, U(1)_{\hat{R}} USp(2Q)^2 = -\frac{1+Q}{2}, \quad \text{Tr}\, U(1)_c USp(2Q)^2 = -1, \quad \text{Tr}\, U(1)_t USp(2Q)^2 = \frac{1-Q}{2}. \tag{60}$$

Then we have cubic anomalies involving a single symmetry

$$\text{Tr}\, U(1)_{\hat{R}}^3 = -Q(1 + 2Q), \qquad \text{Tr}\, U(1)_t^3 = 1 + Q - 2Q^2, \qquad \text{Tr}\, U(1)_c^3 = -8Q. \tag{61}$$

Finally we have cubic anomalies involving several $U(1)$ symmetries

$$\text{Tr}\, U(1)_{\hat{R}} U(1)_t^2 = 0, \qquad \text{Tr}\, U(1)_{\hat{R}} U(1)_c^2 = 0, \qquad \text{Tr}\, U(1)_c U(1)_t^2 = -Q(Q-1), \tag{62}$$

$$\text{Tr}\, U(1)_t U(1)_c^2 = 0, \qquad \text{Tr}\, U(1)_{\hat{R}}^2 U(1)_t = 0, \qquad \text{Tr}\, U(1)_{\hat{R}}^2 U(1)_c = Q(Q+1).$$

To compare with the six dimensional prediction we have to sum the bulk contribution to the inflow contribution of the two punctures with $z = 1/2$, $\xi_G = 1$, $q = -1/2$ and $q_a = -1/2$ for $a = 1, \cdots 8$. For example

$$\text{Tr}\, U(1)_c = \underbrace{-12 \times (1/2) \times 1\, Q}_{\text{geometric}} + \underbrace{2 \times 8 \times (-1/2) Q}_{\text{inflow}} = -14Q,$$

$$\text{Tr}\, U(1)_c^3 = \underbrace{-12 \times (1/2) \times 1^2\, Q}_{\text{geometric}} + \underbrace{2 \times 8 \times (-1/2)^3 Q}_{\text{inflow}} = -8Q, \tag{63}$$

and farther, taking contributions from the punctures only,

$$\text{Tr}\, U(1)_L = -\frac{2Q^2 - Q - 1}{2}$$

$$\text{Tr}\, U(1)_L^3 = -\frac{2Q^2 - Q - 1}{8}$$

$$\text{Tr}(U(1)_c SU(2)_L^2) = -\frac{Q(Q-1)}{4}. \tag{64}$$

In order to match the anomalies (64) with the ones computed in $4d$ we also need to redefine $q_t \to \frac{1}{2} q_t \equiv q_{\hat{t}}$. In this normalization for the $U(1)_{\hat{t}}$ charges the character of the fundamental representation of $SU(2)_L$ is $\hat{t}^{\frac{1}{2}} + \hat{t}^{-\frac{1}{2}}$ and thus $\text{Tr}\, U(1) SU(2)_L^2 = \text{Tr}\, U(1) U(1)_{\hat{t}}^2$. In particular,

$$\text{Tr}\, U(1)_{\hat{t}} = \frac{1}{2} \text{Tr}\, U(1)_t = -\frac{2Q^2 - Q - 1}{2}$$

$$\text{Tr}\, U(1)_{\hat{t}}^3 = \frac{1}{8} \text{Tr}\, U(1)_t^3 = -\frac{2Q^2 - Q - 1}{8}$$

$$\text{Tr}\, U(1)_c U(1)_{\hat{t}}^2 = \frac{1}{4} \text{Tr}\, U(1)_c U(1)_t^2 = -\frac{Q(Q-1)}{4}. \tag{65}$$

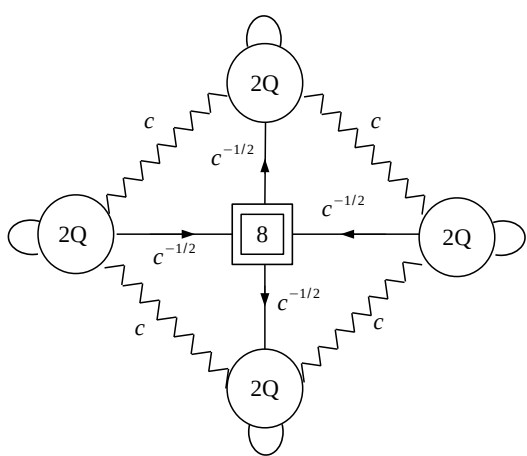

Figure 13: Gluing $2n$ $(\frac{1}{2}, \frac{1}{2}, \frac{1}{2}, \frac{1}{2}, \frac{1}{2}, \frac{1}{2}, \frac{1}{2}, \frac{1}{2})$ tubes we obtain a torus with $n$ units of flux preserving $SU(2)_t \times E_7 \times U(1)$. Here we have $n = 2$.

## 4.3 Tori with $\mathcal{F} = (n, n, n, n, n, n, n, n)$

The simplest tori models we can build are obtained by combining the basic $E_7$ tubes together with a trivial action of the Weyl group. Taking an even number of such tubes we do not break any of the symmetries. In particular combining $2n$ tubes we obtain the torus compactification of the E-string with $n$ units of flux for the $U(1)$ whose commutant in $E_8$ is $E_7$ (see Figure 13).

**Anomalies**:

We compute some of the anomalies of this torus theory. It is convenient to package the anomalies of abelian symmetries into trial $a$ and $c$ anomalies. Using the trial R-charge $R_0 + \mathfrak{t}q_t + \mathfrak{c}q_c$ we first calculate the trial $a$ and $c$ anomalies of $E[USp(2Q)]$

$$
a^{E[USp(2Q)]}(\mathfrak{c}, \mathfrak{t}) = \frac{3}{32}Q\big(-12\mathfrak{c}^3 + \mathfrak{c}(16 - 9(Q-1)(\mathfrak{t}-2)\mathfrak{t}) - (2Q-1)\mathfrak{t}(3(\mathfrak{t}-3)\mathfrak{t}+8) - 4\big) +
$$
$$
- \frac{3}{32}\big(3(1-\mathfrak{t})^3 - (1-\mathfrak{t})\big), \tag{66}
$$
$$
c^{E[USp(2Q)]}(\mathfrak{c}, \mathfrak{t}) = \frac{1}{32}Q\big(-36\mathfrak{c}^3 + \mathfrak{c}(44 - 27(Q-1)(\mathfrak{t}-2)\mathfrak{t}) - (2Q-1)\mathfrak{t}(9(\mathfrak{t}-3)\mathfrak{t}+22) - 8\big) +
$$
$$
- \frac{1}{32}\big(9(1-\mathfrak{t})^3 - 5(1-\mathfrak{t})\big).
$$

When we glue the tubes to a torus we add an octet of fundamental fields $\Phi_j$, the antisymmetric field $\hat{A}$, and gauge the $USp(2Q)$ symmetry. The contribution of the gluing to the anomaly is then,

$$
a^{glue\,(8,0)}(\mathfrak{c}, \mathfrak{t}) = \frac{3}{32}\big(-6\mathfrak{c}^3 n + 8\mathfrak{c}n - (n(2n-1)-1)\big(-3(\mathfrak{t}-1)^3 + \mathfrak{t}-1\big) + 2(2n+1)n\big),
$$
$$
c^{glue\,(8,0)}(\mathfrak{c}, \mathfrak{t}) = \frac{1}{32}\big(2\mathfrak{c}\big(20 - 9\mathfrak{c}^2\big)n + \big(2n^2 - n - 1\big)\big(9\mathfrak{t}^3 - 27\mathfrak{t}^2 + 22\mathfrak{t} - 4\big) + 4(2n+1)n\big).
$$
$$
\tag{67}
$$

Here the label $(8, 0)$ denotes the fact that we glue with an octet of $\Phi_i$ as opposed to gluing with less than 8 fields, as we do when we consider a non-trivial identification with the action

of Weyl symmetry group. The total anomaly is given by

$$a^{E_7,2n}(\mathfrak{c},\mathfrak{t}) = 2n(a^{E[USp(2Q)]}(\mathfrak{c},\mathfrak{t}) + a^{glue\,(8,0)}(\mathfrak{c},\mathfrak{t})),$$
$$a^{E_7,2n}(\mathfrak{c},\mathfrak{t}) = 2n(c^{E[USp(2Q)]}(\mathfrak{c},\mathfrak{t}) + c^{glue\,(8,0)}(\mathfrak{c},\mathfrak{t})).$$

$$(68)$$

We can maximize $a^{E_7,2n}$ with respect to $\mathfrak{c}$ and $\mathfrak{t}$ and obtain,

$$\mathfrak{c} = \frac{2\sqrt{2}}{3}\sqrt{3Q+5}, \qquad \mathfrak{t} = 0, \tag{69}$$

for which we get,

$$a = \frac{\sqrt{2}Q(3Q+5)^{\frac{3}{2}}}{16}z, \qquad c = \frac{\sqrt{2(3Q+5)}Q(3Q+7)}{16}z, \tag{70}$$

which matches the six dimensional prediction (2) with $z = n$ and $\xi_G = 1$, the value corresponding to flux preserving $U(1) \times E_7$. We can also match the abelian anomalies. For example from the $4d$ theory we extract

$$\text{Tr}\,U(1)_c = -12Qn, \qquad \text{Tr}\,U(1)_c^3 = -12Qn, \tag{71}$$

which perfectly match the $6d$ prediction.

**Index**:

Next we can compute the index of the torus theory and check whether the expected symmetry makes an appearance. The index for the basic tube theory with flux $\mathcal{F} = \left(\frac{1}{2}, \frac{1}{2}, \frac{1}{2}, \frac{1}{2}, \frac{1}{2}, \frac{1}{2}, \frac{1}{2}, \frac{1}{2}\right)$ is given by

$$\mathcal{I}_{tube}^{\left(z=\frac{1}{2}\right)}(\vec{x},\vec{y},c,t,\mathbf{u}) = \prod_{n=1}^{Q}\prod_{i=1}^{8}\Gamma_e\left((pq)^{\frac{1}{2}}c^{-\frac{1}{2}}u_i x_n^{\pm 1}\right)\Gamma_e\left((pq)^{\frac{1}{2}}c^{-\frac{1}{2}}u_i^{-1}y_n^{\pm 1}\right)\mathcal{I}_{E[USp(2Q)]}(x_n,y_n,c,t). \tag{72}$$

We also define the contribution of the gluing as,

$$\Delta_Q(\vec{z};\mathbf{u},c,t) = \prod_{n=1}^{Q}\prod_{i=1}^{8}\Gamma_e\left((qp)^{\frac{1}{2}}c^{\frac{1}{2}}u_i^{-1}z_n^{\pm 1}\right)\Delta_Q(z,t),$$

where the contribution of the vector and the traceless antisymmetric of $USp(2Q)$ is

$$\Delta_Q(z,t) = \frac{\Gamma_e(t)^{Q-1}\prod_{n<m}^{Q}\Gamma_e\left(t\,z_n^{\pm 1}z_m^{\pm 1}\right)}{\prod_{n=1}^{Q}\Gamma_e\left(z_n^{\pm 2}\right)\prod_{n<m}^{Q}\Gamma_e\left(z_n^{\pm 1}z_m^{\pm 1}\right)}. \tag{73}$$

Then the index of the torus with $z = n$ units of flux has the following index,

$$\mathcal{I}_Q^{(z=n)} = \oint \frac{d\vec{z}_Q^{(1)}}{2\pi i \vec{z}_Q^{(1)}}\cdots\oint\frac{d\vec{z}_Q^{(2n)}}{2\pi i \vec{z}_Q^{(2n)}}\prod_{i=1}^{2n}\mathcal{I}_{tube}^{\left(z=\frac{1}{2}\right)}(\vec{z}^{(i)},\vec{z}^{(i+1)},c,t,\mathbf{u})\Delta_Q(\vec{z}^{(i+1)};\mathbf{u},c,t). \tag{74}$$

In order to analyze the symmetries of the theory we expand the index using the $6d$ R-symmetry $\hat{R}$. The case of $Q = 1$ was discussed in detail in [1], here we give the result for $Q = 2$ and generic value of $z$ [9]. With the $6d$ R-symmetry $\hat{R}$ and with $t = \hat{t}^{\frac{1}{2}}$ we obtain for flux $z > 1$

$$\mathcal{I}_{Q=2}^{(z=n)} = 1 + c^{2z} + c^{4z} + \cdots + qp(z\mathbf{56}c^{-1} - z\mathbf{56}c + 2zc^{-2}) + \tag{75}$$

$$qp(q+p)(z\mathbf{56}c^{-1} + 2zc^{-2}) + (qp)^{\frac{3}{2}}(z\mathbf{56}c^{-1} + 2zc^{-2})(\hat{t}^{\frac{1}{2}} + \hat{t}^{-\frac{1}{2}}) + \cdots.$$

---

[9]For low values of flux there can be additional operators with low charges contributing in low orders of the expansion of the index.

We see that the representations of $SU(8)$ enhance to $E_7$. In particular $\overline{\mathbf{28}} + \mathbf{28} \to \mathbf{56}$. The fugacity $\hat{t}^{\frac{1}{2}}$ is the Cartan of $SU(2)_L$. We can easily identify some of these operators in the quiver. For example, the operators charged $c^{2z}$ are $\prod_{i=1}^{2n} \Pi_{(i)}$ where $\Pi_{(i)}$ is the $\Pi$ operator defined as in (19) for the $i$th $E[USp(2Q)]$ block. The operators in the $\mathbf{28}$ and $\overline{\mathbf{28}}$ are built from the octects fields as $\mathrm{Tr}\, M_i M_i$. Note as half of $M_i$ are in fundamental and half in antifundamental of $SU(8)$ we get exactly $n$ $\mathbf{56}$s.

We will now compare the $4d$ spectrum that we see from the index with what we expect from the $6d$ construction. It is expected [57] (see also [58] and appendix E of [1]) that the lowest BPS operators contributing to the index of the $4d$ theory come from $6d$ conserved currents and the energy momentum tensor. Since we are considering torus compactifications with flux breaking $E_8 \to U(1)_c \times E_7$ we expect that the contribution to the index of these operators will be in the representations appearing in the branching rule for the decomposition of the adjoint of $E_8 \to U(1)_c \times E_7$:

$$\mathbf{248} \to \mathbf{1}^{\pm 2} \oplus \mathbf{1}^0 \oplus \mathbf{133}^0 \oplus \mathbf{56}^{\pm 1}, \tag{76}$$

where the subscripts indicate the $U(1)_c$ charges. The multiplicity on which these operator contribute depends on the charges and flux $z$. For example an operator with charges $+1$ under the symmetry will be a fermion and contribute with multiplicity $-z$ to the index, while an operator charged $-2$ will be a boson and will contribute with multiplicity $+2z$ to the index. These operators will appear at order $pq$ in the expansion of the $4d$ index when the $6d$ R-charge $\hat{R}$ is chosen. This is the expected pattern. Indeed we see that in (75) at order $pq$ we have operators in the $\mathbf{56}^\pm$ and $\mathbf{1}^2$. However, we are missing the $\mathbf{1}^{-2}$ operator. It is not clear what eliminates it from the 4d theory, and it will be interesting to figure this out. One possibility is that it get canceled against defect operators wrapping the torus.

We then can think of $0 = \mathbf{1}^0 + \mathbf{133}^0 + \mathbf{3}^0_{SU(2)_L} - (\mathbf{1}^0 + \mathbf{133}^0 + \mathbf{3}^0_{SU(2)_L})$ as the cancellation of marginal operators and conserved currents. So we would conclude that the conformal manifold, having dimension 8, is big enough to accommodate the $E_7$ symmetry enhancement.

Another property that we immediately see is that the index is invariant under exchange of $\hat{t}$ with $\hat{t}^{-1}$. This is consistent with the expectation that $\hat{t}^{\frac{1}{2}}$ is an $SU(2)_L$ fugacity as the operation of flipping $\hat{t}$ is the Weyl operation of the $SU(2)$. This property follows directly from the flip-flip duality of $E[USp(2Q)]$ and in particular from (30). Note that, as we always introduce the antisymmetric field $\hat{A}$ for each gluing (30), this will guarantee that the index is invariant under the Weyl transformation of $SU(2)_L$ for tori with any choice of flux. This is not true for tubes however as the punctures break the $SU(2)_L$ symmetry to $U(1)_{\hat{t}}$.

## 4.4 Tori with $\mathcal{F} = \left(\frac{k}{2}, \frac{k}{2}, \frac{k}{2}, \frac{k}{2}, \frac{k}{2}, \frac{k}{2}, \frac{k}{2}, \frac{k}{2}\right)$

If we glue odd number of basic tubes we obtain theories with half-integer fluxes $\mathcal{F} = \left(\frac{k}{2}, \frac{k}{2}, \frac{k}{2}, \frac{k}{2}, \frac{k}{2}, \frac{k}{2}, \frac{k}{2}, \frac{k}{2}\right)$, with $k$ an odd integer. This choice will in general break the $E_7$ symmetry to $U(1)^4$, but we may expect this to further enhance at most to $F_4$ [1].

Tori with half-integer fluxes can be obtained by gluing an odd number of copies of the tube. We focus on the case of a single tube self-glued to give a torus with $\mathcal{F} = \left(\frac{1}{2}, \frac{1}{2}, \frac{1}{2}, \frac{1}{2}, \frac{1}{2}, \frac{1}{2}, \frac{1}{2}, \frac{1}{2}\right)$. Recall that the two punctures of the tube are of types that are not related by an action of the Weyl group of $SO(16)$. Hence, we can't perform a gluing that preserves the $SU(8)$ symmetry, as expected. Instead, we will perform a gauging that explicitly breaks the $SU(8)$ symmetry to $SU(4) \times U(1)$, which actually enhances to $SO(8)$ in the Lagrangian and then discuss the possibility for this to further enhance to $F_4$.

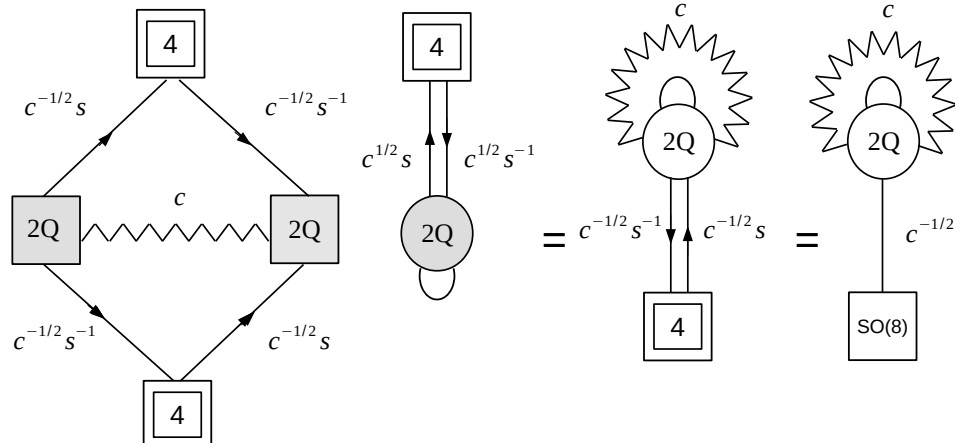

Figure 14: Self-gluing of the minimal tube yields the torus with flux $\left(\frac{1}{2}, \frac{1}{2}, \frac{1}{2}, \frac{1}{2}, \frac{1}{2}, \frac{1}{2}, \frac{1}{2}, \frac{1}{2}\right)$. The shaded nodes indicate the gauging which identifies the $USp(2Q)$ symmetries of the basic tube.

More precisely, we start from the tube theory of Figure 9 and we split the octets of chiral fields in two groups, as depicted on the left of Figure 14. This corresponds to the group decomposition $SU(8)_u \rightarrow SU(4)_v \times SU(4)_w \times U(1)_s$. We also rewrite the superpotential as

$$\mathcal{W} = \sum_{a=1}^{4} M^a \Pi M'_a + \sum_{i=5}^{8} M^i \Pi M'_i. \tag{77}$$

We perform a gauging that breaks the upper $SU(4)$ symmetry by identifying the two $USp(2Q)$ symmetries and adding a pair $\Phi$, $\Phi'$ of bifundamental and anti-bifundamental of $SU(4) \times USp(2Q)$ and an $USp(2Q)$ traceless antisymmetric $\hat{A}$ with superpotential

$$\mathcal{W} = \sum_{i=1}^{4} M^i \Pi M'_i + \sum_{a=5}^{8} M^a \Pi M'_a + \sum_{a=5}^{8} \left(M^a \Phi_a + M'_a \Phi'^a\right) + \hat{A}(A_x - A_y). \tag{78}$$

Integrating out the massive fields we get the quiver on the right of Figure 14 with superpotential

$$\mathcal{W} = \sum_{i=1}^{4} M^i \Pi M'_i, \tag{79}$$

where the global symmetry is actually $SO(8)$. This is the theory corresponding to a torus with flux $\mathcal{F} = \left(\frac{1}{2}, \frac{1}{2}, \frac{1}{2}, \frac{1}{2}, \frac{1}{2}, \frac{1}{2}, \frac{1}{2}, \frac{1}{2}\right)$.

In order to discuss the possible enhancement of the $SO(8)$ symmetry to $F_4$, we consider the superconformal index of this theory

$$\mathcal{I}_{torus}^{\left(z=\frac{1}{2}\right)}(c, t, u_a) = \oint d\vec{z}_Q \, \Delta_Q(z, t) \prod_{n=1}^{Q} \prod_{a=5}^{8} \Gamma_e\left((pq)^{\frac{1}{2}} c^{\frac{1}{2}} \left(s^{-1} w_a\right)^{\pm 1} z_n^{\pm 1}\right) \mathcal{I}_{tube}^{\left(z=\frac{1}{2}\right)}(z_n, z_n, c, t, u_i) =$$

$$= \oint d\vec{z}_Q \, \Delta_Q(z, t) \prod_{n=1}^{Q} \prod_{i=1}^{4} \Gamma_e\left((pq)^{\frac{1}{2}} c^{-\frac{1}{2}} \left(s \, v_i\right)^{\pm 1} z_n^{\pm 1}\right) \mathcal{I}_{E[USp(2Q)]}(z_n, z_n, c, t),$$

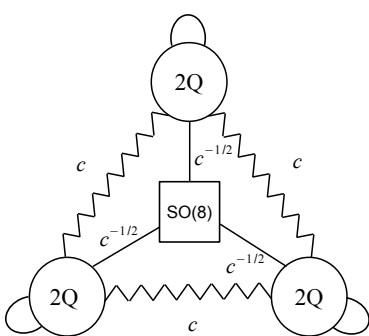

Figure 15: Gluing three basic tubes to form a tube with flux $\left(\frac{3}{2},\frac{3}{2},\frac{3}{2},\frac{3}{2},\frac{3}{2},\frac{3}{2},\frac{3}{2},\frac{3}{2}\right)$.

where we decomposed the $SU(8)_u$ fugacities into $SU(4)_v \times SU(4)_w \times U(1)_s$ fugacities according to

$$u_i = s\, v_i, \qquad u_a = s^{-1} w_a, \tag{80}$$

with the constraints $\prod_{i=1}^4 v_i = \prod_{a=5}^8 w_a = 1$. Notice that the index is manifestly $SO(8)$ invariant in the variables $u_i = s\,v_i$. It is also secretly $F_4$ invariant, since according to Theorem 3.22 of [32] it is invariant under

$$u_a \rightarrow \frac{u_a}{\sqrt{u_1 u_2 u_3 u_4}}\,. \tag{81}$$

This implies that if we expand the index in powers of $p$ and $q$, the characters of $SO(8)$ should actually re-arrange into characters of $F_4$. Indeed, using the $6d$ R-charge $\hat{R}$ and rescaling $t = \hat{t}^{\frac{1}{2}}$ we find

$$\mathcal{I}_{torus}^{\left(z=\frac{1}{2}\right)} = 1 + c + 2c^2 + \cdots + qp\left(\mathbf{28} + 1 + c^{-2} + (\mathbf{28}+1)c^{-1} + c\right) + \tag{82}$$
$$qp(p+q)\left(\mathbf{28} + 2 + c^{-2} + (\mathbf{28}+1)c^{-1} + (\mathbf{28}+2)c\right) +$$
$$(qp)^{\frac{3}{2}}\left(c^{-2} + \mathbf{28}c^{-1} - \mathbf{28}c\right)\left(\hat{t}^{\frac{1}{2}} + \hat{t}^{-\frac{1}{2}}\right) + \cdots,$$

where $\mathbf{28}$ is the representation of $SO(8)$, which can also be thought of as the representations $\mathbf{26} + 1 + 1$ of $F_4$. From this expression we can see that if we compute the index with the $4d$ superconformal R-charge we get a $qp$ term equal to $(\mathbf{28}+1)qp$, which doesn't contain a conserved current for $F_4$. If we assume a cancellation of the current due to marginal operators, we find that the conformal manifold is bigger than the one predicted from $6d$. The expansion of the index then can be re-arranged into characters of $F_4$ and there is no a priori contradiction with the conformal manifold having a locus on which the symmetry enhances to $F_4$. This is to be contrasted with the $Q = 1$ case where with minimal flux $z = \frac{1}{2}$ the conformal manifold did not contain an $F_4$ locus [1].

We can also consider the theory corresponding to higher half-integer flux $z > \frac{1}{2}$ (see Figure 15)

$$\mathcal{I}_{torus}^{\left(z=\frac{n}{2}\right)} = 1 + c^{2z} + 2c^{4z} + \cdots + qp\left(2z(\mathbf{28}-1)c^2 - 2z(\mathbf{28}-1)c + 2z\mathbf{28}c^{-1} + 2zc^{-2}\right) +$$
$$+ qp(p+q)\left(2z\mathbf{28}c^{-1} + 2zc^{-2} - 2z(\mathbf{28}-1)c\right) +$$
$$+ (qp)^{\frac{3}{2}}\left(2z\mathbf{28}c^{-1} + 2zc^{-2} - 2z\mathbf{28}c\right)\left(\hat{t}^{\frac{1}{2}} + \hat{t}^{-\frac{1}{2}}\right) + \cdots. \tag{83}$$

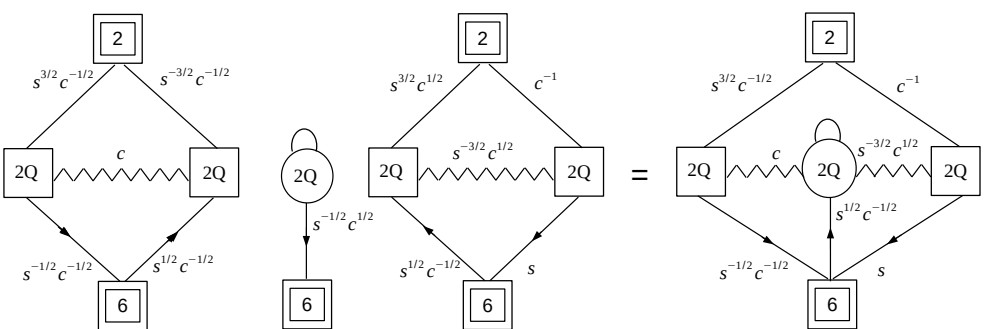

Figure 16: Gluing two basic tubes to form a tube with flux $(1, 1, 1, 1, 1, 1, 0, 0)$. We avoid drawing arrows for lines connecting $USp(2Q)$ nodes to $SU(2)$ nodes.

We notice that in this case there is no $qp$ term corresponding to an operator uncharged under $c$. This means that computing the index with the $4d$ superconformal R-charge the $qp$ term vanishes. Hence, we find no contradiction with the enhancement to $F_4$ on some point of the conformal manifold. Moreover, given that the mixing coefficient with $U(1)_c$ (69) is positive, computing the index with the $4d$ superconformal R-charge there will be no contribution from relevant fermionic operators [59].

### 4.5 Tori with $\mathcal{F} = (2n, 2n, 2n, 2n, 2n, 2n, 0, 0)$

Let us consider gluing two basic tubes with the element of Weyl symmetry group which flips two elements. That is

$$i = 1 \ldots 6 : \quad a_i = b_i, \qquad i = 7, 8 : \quad a_i = 1/b_i. \tag{84}$$

We split the fugacities into $SU(6)_u \times SU(2)_v \times U(1)_s \times U(1)_c$ as,

$$i = 1 \ldots 6 : \quad a_i = s^{\frac{1}{2}} c^{-\frac{1}{2}} u_i, \qquad i = 7, 8 : \quad a_i = s^{-\frac{3}{2}} c^{-\frac{1}{2}} v_i, \tag{85}$$

$$i = 1 \ldots 6 : \quad b_i = s'^{\frac{1}{2}} c'^{-\frac{1}{2}} u'_i, \qquad i = 7, 8 : \quad b_i = s'^{-\frac{3}{2}} c'^{-\frac{1}{2}} v'_i, \tag{86}$$

with $\prod_{i=1}^{6} u_i = \prod_{i=1}^{6} u'_i = \prod_{i=7}^{8} v_i = \prod_{i=7}^{8} v_i = 1$. Then the map between the charges (84) implies $c' = s^{-\frac{3}{2}} c^{\frac{1}{2}}$, $s' = s^{-\frac{1}{2}} c^{-\frac{1}{2}}$, with $u_i = u'_i$ and $v_i = 1/v'_i$.

Since only two fugacities are flipped the gluing will involve six $USp(2Q)$ fundamentals $\Phi_i$, $i = 1, \cdots, 6$ as shown in Figure 16 and flux associated to this tube will be:

$$\left( \frac{1}{2}, \frac{1}{2}, \frac{1}{2}, \frac{1}{2}, \frac{1}{2}, \frac{1}{2}, \frac{1}{2}, \frac{1}{2} \right) + \left( \frac{1}{2}, \frac{1}{2}, \frac{1}{2}, \frac{1}{2}, \frac{1}{2}, \frac{1}{2}, -\frac{1}{2}, -\frac{1}{2} \right) = (1, 1, 1, 1, 1, 1, 0, 0), \tag{87}$$

corresponding to half a unit of flux $z = \frac{1}{2}$ for the $U(1)$ whose commutant in $E_8$ is $E_6 \times SU(2)$.

If we now glue $2n$ such tubes with a trivial element of the $SO(16)$ Weyl group we obtain the theory corresponding to the compactification on a torus with $z = n$ units of flux in this $U(1)$ and we expect the symmetry on some locus of the conformal manifold to be $SU(2)_L \times E_6 \times SU(2) \times U(1)$. The torus theory is depicted in Figure 17. We proceed to perform some checks of the proposal.

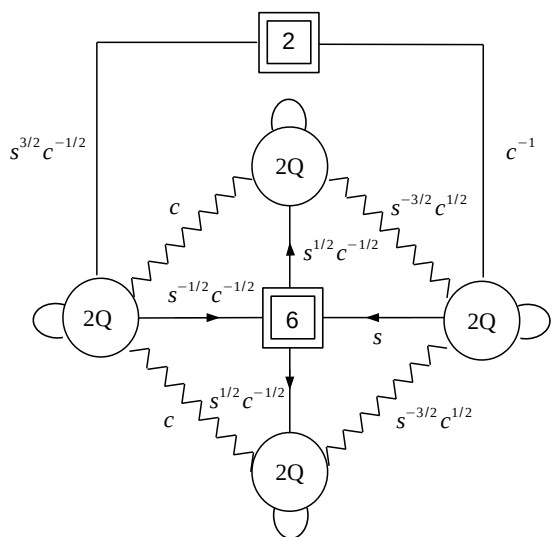

Figure 17: Gluing $2n$ $(1,1,1,1,1,1,0,0)$ tubes we obtain a torus with $n$ units of flux preserving $SU(2)_t \times E_6 \times SU(2) \times U(1)$. Here we have $n = 1$.

**Anomalies**:

We first calculate the conformal anomalies for this torus theory and obtain:

$$a = \frac{1}{8}\sqrt{\frac{3}{2}}zQ(3Q+5)^{3/2}, \qquad c = \frac{1}{8}\sqrt{\frac{3}{2}}zQ\sqrt{3Q+5}(3Q+7). \tag{88}$$

This matches the six dimensional prediction (2) for $SU(2) \times E_6$ preserving $n$ units of flux, that is with $\xi_G = 3$ and $z = n$.

**Index**:

We can compute the index. Again we will consider the case with rank $Q = 2$ and arbitrary flux $z = n$ for simplicity (for the case $Q = 1$ see [1]). Using the six dimensional R-charge $\hat{R}$ and rescaling $t = \hat{t}^{\frac{1}{2}}$ we find

$$
\begin{aligned}
\mathcal{I}^{(z=n)}_{Q=2} =\,& 1 + \cdots + \\
& qp\left(3z(\mathbf{2},\mathbf{1})m^{-3} + 2z(\mathbf{1},\mathbf{27})m^{-2} + z(\mathbf{2},\overline{\mathbf{27}})m^{-1} - 2z(\mathbf{1},\overline{\mathbf{27}})m^2 - z(\mathbf{2},\mathbf{27})m\right) + \\
& qp(q+p)(3z(\mathbf{2},\mathbf{1})m^{-3} + 2z(\mathbf{1},\mathbf{27})m^{-2} + z(\mathbf{2},\overline{\mathbf{27}})m^{-1} - z(\mathbf{2},\mathbf{27})m) + \\
& (qp)^{\frac{3}{2}}(3z(\mathbf{2},\mathbf{1})m^{-3} + 2z(\mathbf{1},\mathbf{27})m^{-2} + z(\mathbf{2},\overline{\mathbf{27}})m^{-1} - z(\mathbf{2},\mathbf{27})m)(\hat{t}^{\frac{1}{2}} + \hat{t}^{-\frac{1}{2}}) + \cdots,
\end{aligned}
\tag{89}
$$

where we redefined the fugacities for the abelian symmetries with respect to the ones used in Figure 16 according to

$$c = m^{\frac{3}{2}}w^{-1}, \qquad s = m^{-\frac{1}{2}}w^{-1}, \tag{90}$$

to isolate $U(1)_w$ which is the one enhancing to $SU(2)$. In (89) we indicate by $(\cdot,\cdot)$ the characters of $SU(2) \times E_6$, for example $(\mathbf{2},\mathbf{1}) = w^2 + w^{-2}$. Then the $SU(2)_v \times SU(6)_u$ fugacities re-organize in the index in terms of characters of $E_6$ according to the branching rules

$$(\mathbf{1},\overline{\mathbf{27}}) = (\mathbf{2},\mathbf{6})_{SU(2)_v \times SU(6)_u} \oplus (\mathbf{1},\overline{\mathbf{15}})_{SU(2)_v \times SU(6)_u}. \tag{91}$$

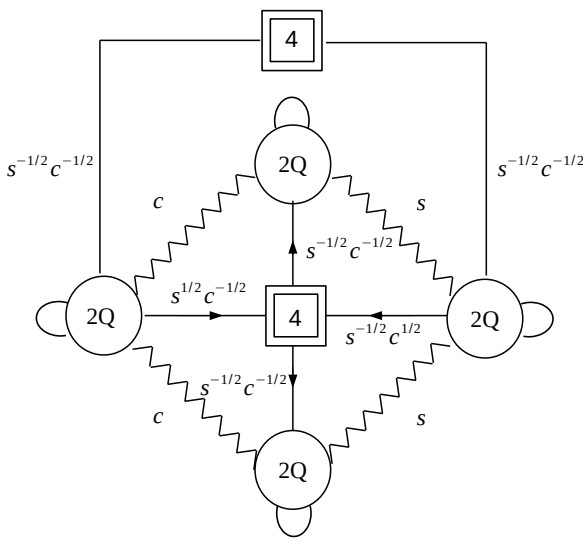

Figure 18: Gluing $2n$ $(1,1,1,1,0,0,0,0)$ tubes we obtain a torus with $z = n$ units of flux preserving $SU(2)_t \times SO(14) \times U(1)$. Here we have $n = 1$.

We can also use the index result to compare with the $6d$ prediction of the spectrum. Since we are considering torus compactifications with flux breaking $E_8 \to U(1)_v \times SU(2) \times E_6$ we expect that the contribution to the index corresponding to the $6d$ conserved currents and energy momentum tensor will appear in the $pq$ term of the index in the representations involved in the branching rule

$$\mathbf{248} \to (\mathbf{1},\mathbf{1})^0 \oplus (\mathbf{3},\mathbf{1})^0 \oplus (\mathbf{1},\mathbf{78})^0 \oplus (\mathbf{1},\mathbf{27})^2 \oplus (\mathbf{1},\overline{\mathbf{27}})^{-2} \oplus (\mathbf{2},\mathbf{27})^{-1} \oplus (\mathbf{2},\overline{\mathbf{27}})^1 \oplus (\mathbf{2},\mathbf{1})^{\pm 3} \,, \tag{92}$$

where the subscripts indicate the $U(1)_v$ charges. Indeed we see that in (89) at order $pq$ we have operators in the $(\mathbf{1},\mathbf{27})^2$, $(\mathbf{1},\overline{\mathbf{27}})^{-2}$, $(\mathbf{2},\mathbf{27})^{-1}$, $(\mathbf{2},\overline{\mathbf{27}})^1$ and $(\mathbf{2},\mathbf{1})^3$. In particular, these appear with a coefficient determined by the value of the flux $z = n$ and their charge under $U(1)_v$. We again can think of

$$0 = (\mathbf{1},\mathbf{1})^0 + (\mathbf{3},\mathbf{1})^0 + (\mathbf{1},\mathbf{78})^0 + \mathbf{3}^0_{SU(2)_L} - \left( (\mathbf{1},\mathbf{1})^0 + (\mathbf{3},\mathbf{1})^0 + (\mathbf{1},\mathbf{78})^0 + \mathbf{3}^0_{SU(2)_L} \right)$$

as the cancellation of marginal operators and $4d$ conserved currents, which is compatible with the dimension of the conformal manifold predicted from $6d$. We are again missing the $(\mathbf{2},\mathbf{1})^{-3}$ operator, and it will be interesting to understand the mechanism causing this.

## 4.6 Tori with $\mathcal{F} = (2n, 2n, 2n, 2n, 0, 0, 0, 0)$

We can glue $2n$ tubes with $(1,1,1,1,0,0,0,0)$ fluxes given in Figure 11, with a trivial action of the $SO(16)$ Weyl group to construct tori with $z = n$ units of flux for the $U(1)$ whose commutant in $E_8$ is $SO(14)$ as shown in Figure 18.
**Anomalies**:
    The conformal anomalies are given by:

$$a = \frac{1}{8} z Q (3Q+5)^{3/2} \,, \qquad c = \frac{1}{8} z Q \sqrt{3Q+5}(3Q+7) \,. \tag{93}$$

This matches the six dimensional prediction (2) for $SO(14)$ preserving $n$ units of flux, that is with $\xi_G = 2$ and $z = n$.

**Index:**

Again we compute the index in the case with rank $Q = 2$ for simplicity (for the case $Q = 1$ see [1]) and generic flux. We use the six dimensional R-charge $\hat{R}$ and rescale $t = \hat{t}^{\frac{1}{2}}$

$$\mathcal{I}_{Q=2}^{(z=n)} = 1 + \cdots + qp(2z\mathbf{14}m^{-2} + z\mathbf{64}m^{-1} - z\mathbf{64}m) + \tag{94}$$

$$qp(q+p)(2z\mathbf{14}m^{-2} + z\mathbf{64}m^{-1}) + (qp)^{\frac{3}{2}}(2z\mathbf{14}m^{-2} + z\mathbf{64}m^{-1})(\hat{t}^{\frac{1}{2}} + \hat{t}^{-\frac{1}{2}}) + \cdots,$$

where we redefined the fugacities for the abelian symmetries with respect to the ones used in Figure 11 according to

$$c = mw^{-1}, \qquad s = mw, \tag{95}$$

which is useful since the $U(1)_w$ symmetry is the one contributing to the enhancement to $SO(14)$ together with the two $SU(4)$ symmmetries. Indeed, the corresponding fugacities re-organize in the index in terms of characters of $SO(14)$ according to the branching rules

$$\mathbf{14} = (\mathbf{1},\mathbf{1})^{\pm 2} \oplus (\mathbf{6},\mathbf{1})^0 \oplus (\mathbf{1},\mathbf{6})^0,$$
$$\mathbf{64} = (\mathbf{4},\overline{\mathbf{4}})^{\pm 1} \oplus (\overline{\mathbf{4}},\mathbf{4})^{\pm 1}. \tag{96}$$

We can also use the index result to compare with the $6d$ prediction of the spectrum. Since we are considering torus compactifications with flux breaking $E_8 \to U(1)_v \times SO(14)$ we expect that the contribution to the index of these operators will be in the representations appearing in

$$\mathbf{248} \to \mathbf{1}^0 \oplus \mathbf{91}^0 \oplus \mathbf{14}^{\pm 2} \oplus \mathbf{64}^- \oplus \overline{\mathbf{64}}^1, \tag{97}$$

where the subscripts indicate the $U(1)_v$ charges. Indeed we see that in (95) at order $pq$ we have operators in the $\mathbf{14}^2$, $\overline{\mathbf{64}}^1$ and $\mathbf{64}^{-1}$. We again can think of

$$0 = \mathbf{1}^0 + \mathbf{91}^0 + \mathbf{3}^0_{SU(2)_L} - \left(\mathbf{1}^0 + \mathbf{91}^0 + \mathbf{3}^0_{SU(2)_L}\right)$$

as the cancellation of marginal operators and conserved currents.

## 4.7 Tori with $\mathcal{F} = (2n, 2n, 0, 0, 0, 0, 0, 0)$ and the braid relation

Let us consider gluing two basic tubes with the element of Weyl symmetry group which flips six elements. That is

$$i = 1, 2: \quad a_i = b_i, \qquad i = 3, \dots 8: \quad a_i = 1/b_i. \tag{98}$$

We split the fugacities into $SU(2) \times SU(6) \times U(1)_s \times U(1)_c$ as,

$$i = 1, 2: \quad a_i = s^{-\frac{3}{2}}c^{-\frac{1}{2}}u_i, \qquad i = 3\dots 8: \quad a_i = s^{\frac{3}{2}}c^{-\frac{1}{2}}v_i, \tag{99}$$

$$i = 1, 2: \quad b_i = s'^{-\frac{3}{2}}c'^{\frac{1}{2}}u'_i, \qquad i = 3\dots 8: \quad b_i = s'^{\frac{3}{2}}c'^{-\frac{1}{2}}v'_i, \tag{100}$$

with $\prod_{i=1}^2 u_i = \prod_{i=1}^2 u'_i = \prod_{i=3}^8 v_i = \prod_{i=3}^8 v'_i = 1$. Then the map between the charges implies $c' = s^{\frac{3}{2}}c^{-\frac{1}{2}}$, $s' = s^{\frac{1}{2}}c^{\frac{1}{2}}$, with $u_i = u'_i$ and $v_i = 1/v'_i$. Now six fugacities are flipped so the gluing will involve only two $USp(2Q)$ fundamentals $\Phi_i$, $i = 1, 2$, as shown in Figure 19.

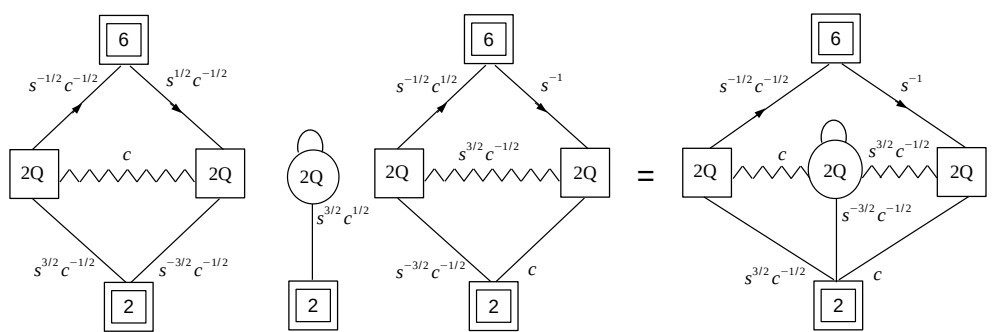

Figure 19: Gluing two basic tubes to form a tube with flux $(1, 1, 0, 0, 0, 0, 0, 0)$.

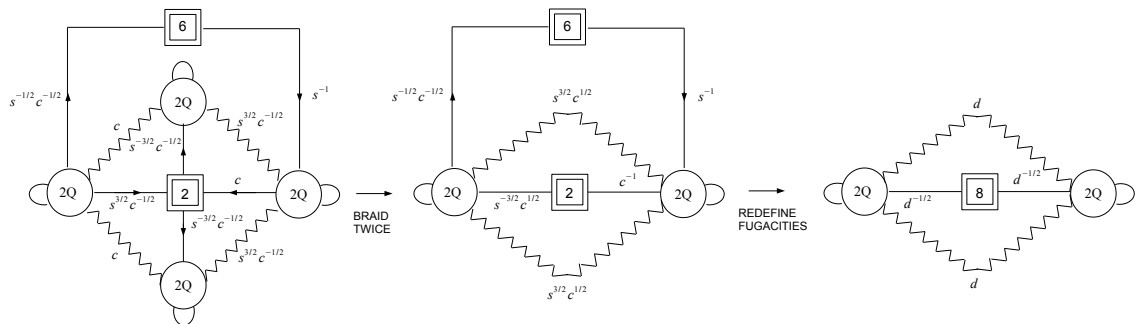

Figure 20: On the lhs the torus with flux $(2, 2, 0, 0, 0, 0, 0, 0)$. We apply twice the braid relation to obtain the quiver in the middle. Finally by redefining the fugacities we obtain the quiver on the rhs corresponding to flux $(1, 1, 1, 1, 1, 1, 1, 1)$.

Note that the flux is

$$\left(\frac{1}{2}, \frac{1}{2}, \frac{1}{2}, \frac{1}{2}, \frac{1}{2}, \frac{1}{2}, \frac{1}{2}, \frac{1}{2}\right) + \left(\frac{1}{2}, \frac{1}{2}, -\frac{1}{2}, -\frac{1}{2}, -\frac{1}{2}, -\frac{1}{2}, -\frac{1}{2}, -\frac{1}{2}\right) = (1, 1, 0, 0, 0, 0, 0, 0). \tag{101}$$

This is also an $E_7$ flux related to $(\frac{1}{2}, \frac{1}{2}, \frac{1}{2}, \frac{1}{2}, \frac{1}{2}, \frac{1}{2}, \frac{1}{2}, \frac{1}{2})$ by the action of the Weyl symmetry (see appendix A). Thus in order for the picture to be consistent torus theories obtained by gluing either type of tubes have to be equivalent. This is indeed the case due to the braid relation discussed in section 3.4.

For example, as shown in Figure 20, if we glue two $(1, 1, 0, 0, 0, 0, 0, 0)$ tubes and apply twice the braid relation we obtain the torus with $(1, 1, 1, 1, 1, 1, 1, 1)$ flux, provided we redefine the fugacities as

$$v_i = \tilde{v}_i s^{-1/4} c^{1/4}, \qquad u_j = \tilde{u}_j s^{3/4} c^{-3/4}, \tag{102}$$

which recombine into the $SU(8)$ fugacities $x_i = \tilde{u}_i$ for $i = 1, 2$ and $x_i = \tilde{v}_i$ for $i = 3, \cdots 8$ satisfying $\prod_{i=1}^{8} x_i = 1$, and we define the remaining $U(1)$ fugacity $d = s^{3/2} c^{1/2}$.

**Anomalies:**

Gluing $2n$ tubes together we can easily compute the conformal anomalies and obtain that

they give us,

$$a = \frac{1}{8}\sqrt{\frac{1}{2}z}Q(3Q+5)^{3/2}, \qquad c = \frac{1}{8}\sqrt{\frac{1}{2}z}Q\sqrt{3Q+5}(3Q+7).\qquad (103)$$

This matches the six dimensional prediction (2) for $E_7$ preserving $n$ units of flux, that is with $\xi_G = 1$ and $z = n$.

**Index**: As we have seen the braid relation (37) guarantees that the index of this theory is the same as the one of the $E_7$ torus.

## 5 Flowing to $3d$

In this section we study the dimensional reduction to $3d$ of $E[USp(2Q)]$ and various real mass flows which connect it to well known $3d$ theories.

**Compactification to $T[USp(2Q)]$**

If we compactify the $E[USp(2Q)]$ theory on a circle we obtain a $3d$ $\mathcal{N} = 2$ quiver theory we denote as $T[USp(2Q)]$ which has the same gauge and matter content and superpotential

$$\mathcal{W}_{TUSp(2Q)} = \mathcal{W}_{E[USp(2Q)]} + \mathcal{W}_{mon}, \qquad (104)$$

where $\mathcal{W}_{mon}$ is the contribution of KK monopoles turned on for each gauge node which are generated in the reduction as in [60]. This monopole superpotential ensures that the $T[USp(2Q)]$ and $E[USp(2Q)]$ have the same global symmetry,

$$USp(2Q)_M \times USp(2Q)_T \times U(1)_{m_A} \times U(1)_\Delta, \qquad (105)$$

since the condition of marginality of the $USp(2n)$ monopoles in $3d$ is equivalent to the requirement that $U(1)_R$ is non-anomalous in $4d$. The gauge invariant operators of $T[USp(2Q)]$ can be constructed in the same way as those of $E[USp(2Q)]$ since the monopole superpotential also implies that the monopole operators are not in the chiral ring.

We can implement the $3d$ limit on the $\mathbb{S}^3 \times \mathbb{S}^1$ supersymmetric index [60–62] by rescaling the global and gauge fugacities with the $\mathbb{S}^1$ radius $r$ as

$$x_n \to e^{2\pi i r M_n}, \quad y_n \to e^{2\pi i r T_n}, \quad c \to e^{2\pi i r \Delta}, \quad t \to e^{2\pi i r(i\omega - 2m_A)}, \quad z_\alpha^{(i)} \to e^{2\pi i r z_\alpha^{(i)}}, \quad (106)$$

where $i = 1, \cdots, Q-1$ and $\alpha = 1, \cdots, i$ and taking the hyperbolic limit of the elliptic Gamma function:

$$\lim_{r \to 0} \Gamma_e\left(e^{2\pi i r x}; e^{-2\pi r b}, e^{-2\pi r b^{-1}}\right) = e^{\frac{i\pi}{6r}\left(i\frac{\omega}{2} - x\right)} s_b\left(i\frac{\omega}{2} - x\right), \qquad (107)$$

with $\omega = b + b^{-1}$. By doing so we find:

$$\lim_{r \to 0} \mathcal{I}_{E[USp(2Q)]}(x_n, y_n, t, c) = C_Q^{3d}(m_A, \Delta, r)\mathcal{Z}_{T[USp(2Q)]}(M_n, T_n, m_A, \Delta), \qquad (108)$$

where

$$
\mathcal{Z}_{T[USp(2Q)]}(M_n, T_n, m_A, \Delta) = s_b\left(-i\frac{\omega}{2} + 2\Delta\right) s_b\left(i\frac{\omega}{2} - 2m_A\right)^{Q-1} \times
$$

$$
\times \prod_{n<m}^{Q} s_b\left(i\frac{\omega}{2} \pm M_n \pm M_m - 2m_A\right) \prod_{n=1}^{Q} s_b\left(i\frac{\omega}{2} \pm T_Q \pm M_n - \Delta\right) \times
$$

$$
\times \int d\vec{z}_{Q-1} \frac{\prod_{i=1}^{Q-1} s_b\left(\pm T_Q \pm z_i - m_A + \Delta\right) \prod_{n=1}^{Q} s_b\left(\pm z_i \pm M_n + m_A\right)}{s_b\left(-i\frac{\omega}{2} + 2m_A\right) \prod_{i<j}^{Q-1} s_b\left(i\frac{\omega}{2} \pm z_i \pm z_j\right) \prod_{i=1}^{Q-1} s_b\left(i\frac{\omega}{2} \pm 2z_i\right)} \times
$$

$$
\times \mathcal{Z}_{T[USp(2(Q-1))]}\left(z_1, \cdots, z_{Q-1}, T_1, \cdots, T_{Q-1}, m_A, \Delta + m_a - i\frac{\omega}{2}\right), \tag{109}
$$

is the partition function of the $T[USp(2Q)]$ theory on $\mathbb{S}_b^3$ [63–66]. The integration measure is defined now as

$$
d\vec{z}_n = \frac{1}{2^n n!} \prod_{i=1}^{n} dz_i, \tag{110}
$$

and the prefactor is

$$
C_Q^{3d}(m_A, \Delta, r) = \left[ r(e^{-2\pi rb}; e^{-2\pi rb})_\infty (e^{-2\pi rb^{-1}}; e^{-2\pi rb^{-1}})_\infty \right]^{\frac{Q(Q-1)}{2}} \times
$$

$$
\times e^{\frac{i\pi}{6r}\left(Q(9Q-1)i\frac{\omega}{4} - 2Q(2Q-1)m_A - 2Q\Delta\right)}. \tag{111}
$$

For example explicitly in the $Q = 2$ case we get

$$
\lim_{r\to 0} \mathcal{I}_{E[USp(4)]}(x_n, y_n, t, c) = C_2^{3d} s_b\left(-i\frac{\omega}{2} + 2\Delta\right) s_b\left(-\frac{3}{2}i\omega + 2m_A + 2\Delta\right) \times
$$

$$
\times s_b\left(i\frac{\omega}{2} - 2m_A\right)^2 s_b\left(i\frac{\omega}{2} \pm M_1 \pm M_2 - 2m_A\right) \prod_{n=1}^{2} s_b\left(i\frac{\omega}{2} \pm T_2 \pm M_n\right) \times
$$

$$
\times \int dz_1 \frac{s_b\left(i\omega \pm z \pm T_1 - m_A - \Delta\right) \prod_{n=1}^{2} s_b\left(\pm z \pm M_n + m_A\right)}{s_b\left(i\frac{\omega}{2} \pm 2z\right) s_b\left(\pm z \pm T_2 + m_A - \Delta\right)} =
$$

$$
= C_2^{3d} \mathcal{Z}_{T[USp(4)]}(M_n, T_n, m_A, \Delta), \tag{112}
$$

where now the integration measure is

$$
dz_1 = \frac{dz}{2}. \tag{113}
$$

The divergent prefactor is in this case

$$
C_2^{3d} = r(e^{-2\pi rb}; e^{-2\pi rb})_\infty (e^{-2\pi rb^{-1}}; e^{-2\pi rb^{-1}})_\infty e^{\frac{i\pi}{6r}\left(\frac{17}{2}i\omega - 12m_A - 4\Delta\right)}. \tag{114}
$$

It is then easy to use the recursive definition of $E[USp(2Q)]$ and of $T[USp(2Q)]$ to obtain (108), (109).

Notice that $T[USp(2Q)]$ inherits the dualities of the mother $E[USp(2Q)]$ theory. In particular it is self-dual under the duality swapping the two $USp(2N)$ groups. Indeed we see that if we take the $3d$ limit of the self-duality identity (26) the divergent prefactor $C_N^{3d}(m_A, \Delta, r)$, which is independent from $M_a$ and $T_a$, cancels out yielding the self-duality identity for $T[USp(2Q)]$

$$
\mathcal{Z}_{T[USp(2Q)]}(M_n, T_n, m_A, \Delta) = \mathcal{Z}_{T[USp(2Q)]}(T_n, M_n, m_A, \Delta). \tag{115}
$$

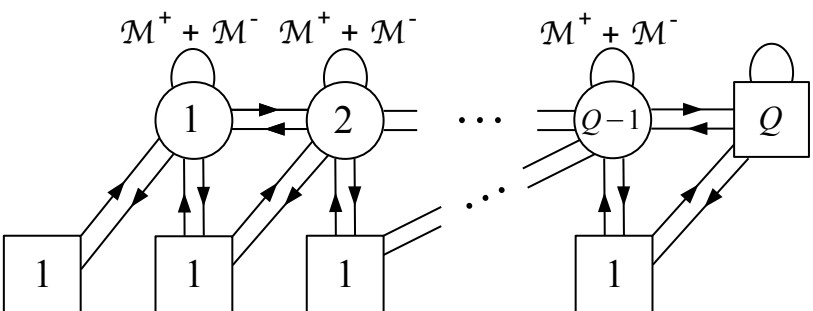

Figure 21: Quiver diagram of the three-dimensional $FM[U(Q)]$ theory. Both square and circular nodes denote $U(n)$ symmetries. Double-lines connecting two nodes represent pairs of bifundamental chirals in conjugate representations with respect to the corresponding symmetries. Lines that start and end on the same node correspond to chirals in the adjoint representation.

**Flow to $FM[SU(Q)]$**

We can now perform some real mass flows to other $3d$ quiver theories. For example we can proceed as in [67] and a perform a real mass deformation that breaks the gauge groups as well as the two $USp(2Q)$ global symmetries to $U(n)$ groups. This can be achieved by considering a real mass deformation of $T[USp(2Q)]$ for the Cartans of the two $USp(2Q)$ global symmetries. All the flavors become massive in the trivial vacuum but we can go to a vacuum far from the origin of the Coulomb branch where each $USp(2n)$ gauge group is broken to $U(n)$ and has $2n + 2$ flavors that remain light.

This flow has the effect of generating non-perturbative contributions due to the breaking of the gauge groups. These contributions together with the original KK monopoles combine in a contribution to superpotential consisting in the sum of the two fundamental monopole operators of opposite magnetic charge at each gauge node $\mathcal{M}^+ + \mathcal{M}^-$. The final theory we reach is the $FM[SU(Q)]$ quiver theory which has been studied recently in [24]. In that context, it was obtained by uplifting free field correlators for $2d$ CFT, following the strategy discussed in [45]. Here instead we found a $4d$ origin of the $FM[SU(Q)]$ theory.

The content of the theory is specified by the quiver of Figure 21. The superpotential consists of three main parts. One is a cubic superpotential coupling the bifundamentals and the adjoint chirals. The second one is another cubic superpotential, but this time between the chirals in each triangle of the quiver. Finally, we have a monopole superpotential. As a consequence of this superpotential, the global symmetry of the theory is

$$SU(Q)_T \times SU(Q)_M \times U(1)_A \times U(1)_\Delta\,. \tag{116}$$

At the level of the sphere partition function this real mass deformation is implemented [60, 68] by taking

$$M_n \to M_n + s, \quad T_n \to T_n + s, \quad s \to +\infty \tag{117}$$

and by shifting all the integration variables

$$z_\alpha^{(i)} \to z_\alpha^{(i)} + s\,. \tag{118}$$

Notice that for each node since the integrands are symmetric we can rewrite the integrals as:

$$\int_{-\infty}^{+\infty} \prod_{i=1}^{n} dz_i \, f(z_i) = 2^n \int_{0}^{+\infty} \prod_{i=1}^{n} dz_i \, f(z_i) = 2^n \int_{-s}^{+\infty} \prod_{i=1}^{n} dz_i \, f(z_i + s). \tag{119}$$

This has the effect of cancelling the $2^n$ factor in the $USp(2n)$ measure.

The real mass deformation is implemented using

$$\lim_{x \to \pm\infty} s_b(x) = e^{\pm i \frac{\pi}{2} x^2}, \tag{120}$$

and we obtain:

$$\lim_{s \to +\infty} \mathcal{Z}_{T[USp(2Q)]}(M_n + s, T_n + s, m_A, \Delta) = C_Q(M_n, T_n, m_A, \Delta, s) s_b\left(-i\frac{\omega}{2} + 2m_A\right)$$

$$\times \prod_{j=1}^{Q} s_b\left(-i\frac{\omega}{2} + 2\Delta + 2(j-1)(m_A - i\frac{\omega}{2})\right) \mathcal{Z}_{FM[U(Q)]}(M_n, T_n, m_A, \Delta), \tag{121}$$

where the partition function of $FM[U(Q)]$ is

$$\mathcal{Z}_{FM[U(Q)]}(M_n, T_n, m_A, \Delta) = \prod_{n,m=1}^{Q} s_b\left(i\frac{\omega}{2} + (M_n - M_m) - 2m_A\right) \times$$

$$\times \prod_{n=1}^{Q} s_b\left(i\frac{\omega}{2} \pm (M_n - T_Q) - \Delta\right) \int \frac{dx_{Q-1}}{\prod_{i<j}^{Q-1} s_b\left(i\frac{\omega}{2} \pm (x_i^{(Q-1)} - x_j^{(Q-1)})\right)} \times$$

$$\times \prod_{i=1}^{Q-1} s_b\left(\pm(x_i^{(Q-1)} - T_Q) + \Delta - m_A\right) \prod_{n=1}^{Q} s_b\left(\pm(x_i^{(Q-1)} - M_n) + m_A\right) \times$$

$$\times \mathcal{Z}_{FM[U(Q-1)]}\left(x_1^{(Q-1)}, \cdots, x_{Q-1}^{(Q-1)}, T_1, \cdots, T_{Q-1}, m_A, \Delta + m_A - i\frac{\omega}{2}\right), \tag{122}$$

with integration measure

$$dx_k = \frac{1}{k!} \prod_{i=1}^{k} dx_i^{(k)}, \tag{123}$$

while the prefactor is

$$C_Q(M_n, T_n, m_A, \Delta, s) = e^{2\pi i \left(i\frac{\omega}{2} - \Delta + (Q-1)(i\frac{\omega}{2} - m_A)\right)\left(2Qs + \sum_{n=1}^{Q}(M_n + T_n)\right)}. \tag{124}$$

The other prefactors in (122) are flipping fields of the diagonal mesons.

For example explicitly for the $Q = 2$ case we find

$$\lim_{s \to +\infty} \mathcal{Z}_{T[USp(4)]}(M_n + s, T_n + s, m_A, \Delta) = C_2 \, s_b\left(-i\frac{\omega}{2} + 2\Delta\right) s_b\left(-\frac{3}{2}i\omega + 2m_A + 2\Delta\right) \times$$

$$\times s_b\left(i\frac{Q}{2} \pm (M_1 - M_2) - 2m_A\right) \prod_{n=1}^{2} s_b\left(i\frac{\omega}{2} \pm (M_n - T_2) - \Delta\right) s_b\left(i\frac{\omega}{2} - 2m_A\right)^2 \times$$

$$\times \int dx \, s_b\left(i\omega \pm (x - T_1) - \Delta - m_A\right) \times s_b\left(\pm(x - T_2) + \Delta - m_A\right) \prod_{n=1}^{2} s_b\left(\pm(x - M_n) + m_A\right) =$$

$$= C_2 \, s_b\left(-i\frac{\omega}{2} + 2m_A\right) s_b\left(-i\frac{\omega}{2} + 2\Delta\right) s_b\left(-\frac{3}{2}i\omega + 2m_A + 2\Delta\right) \mathcal{Z}_{FM[U(2)]}(M_n, T_n, m_A, \Delta), \tag{125}$$

with

$$C_2 = e^{2\pi i(i\omega - \Delta - m_A)(4s + \sum_n(M_n + T_n))}. \tag{126}$$

Again, one can easily prove that (121) holds for any $Q$ by induction using the recursive definitions (109) and (122).

The partition function of $FM[SU(Q)]$ is obtained from $\mathcal{Z}_{FM[U(Q)]}(M_n, T_n, m_A, \Delta)$ by imposing the constraint on the real masses corresponding to the tracelessness conditions of $SU(N)_M$ and $SU(N)_T$

$$\sum_{n=1}^{Q} M_n = \sum_{n=1}^{Q} T_n = 0. \tag{127}$$

The $FM[SU(Q)]$ has been studied in detail in [24], where in particular it was discussed that it is self-dual under a duality that swaps the two $SU(Q)_M$ and $SU(Q)_T$ symmetries. Here we see a different perspective of this self-duality as a consequence of the self-duality of $T[USp(2Q)]$ which is inherited by $E[USp(2Q)]$. Indeed, we can easily see that by taking the limit (121) on both sides of the $T[USp(2Q)]$ self-duality (115), we get

$$\mathcal{Z}_{FM[SU(Q)]}(M_n, T_n, m_A, \Delta) = \mathcal{Z}_{FM[SU(Q)]}(T_n, M_n, m_A, \Delta), \tag{128}$$

where the prefactors $C_Q$ as well as the contribution of the flipping singlets cancel out between the two sides of the identity since they are symmetric under $M_n \leftrightarrow T_n$.

**Flow to $FT[SU(Q)]$**

As observed in [24] a real mass deformation for the $U(1)_\Delta$ symmetry triggers an RG flow which takes the $FM[SU(Q)]$ to the $FT[SU(Q)]$ theory which is the $T[SU(Q)]$ of Gaiotto–Witten [27] with an extra set of singlets flipping the Higgs branch moment map [26].

It is easy to see indeed when this real mass deformation is turned on all the diagonal and vertical flavors become massive and when interarted out they generate mixed Chern-Simons couplings and restore the topological symmetry at each node lifting the monopole superpotential.

**The Dotsenko–Fateev integral kernel**

As noticed in [24] the $FM[SU(Q)]$ theory is related to yet another interesting object, the kernel function $K_\Delta^{(Q)}(x, y)$ for complex Dotsenko–Fateev (DF) integrals.

Dotsenko-Fateev (DF) integrals appear in the study of $2d$ CFTs as for example in Liouville or Toda theories. When the momenta of the vertex operators in a correlator satisfy the so-called screening condition (meaning that their sum is proportional to an integer $Q$) the correlator in the interacting CFT develops a pole and its residue coincides with a correlator in the free theory (free field) in presence of $Q$ screening charges [69].

The goal is to evaluate the free field DF correlator and perform analytic continuation in $Q$ so to lift the screening condition and reconstruct the correlator with generic momenta in the interacting theory. Typically implementing this procedure is very hard but sometimes this is possible and the kernel function was introduced for this purpose [46, 70]. The kernel function is a complex integral which can be recursively defined as:

$$K_{\Delta}^{(Q)}(x_1, \cdots x_Q, y_1 \cdots y_Q) = \frac{\gamma(-Qb^2)}{\gamma(-b^2)^Q} \prod_{i<j}^{Q} |x_i - x_j|^{2+4b^2} \prod_{k=1}^{Q} |x_k - y_1|^{2\Delta} \times$$

$$\int d\vec{z}_{Q-1}^2 \prod_{i<j}^{Q-1} |z_i - z_j|^2 \prod_{j=1}^{Q-1} |z_j - y_1|^{-2\Delta+2b^2} \prod_{k=1}^{Q} |z_j - x_k|^{-2-2b^2} K_{\Delta+b^2}^{(Q-1)}(z_1, \cdots z_{Q-1}, y_2 \cdots y_Q),$$

$$(129)$$

where

$$d\vec{z}_n^2 = \frac{1}{\pi^n n!} \prod_{i=1}^{n} dz_i^2 \,, \tag{130}$$

and the parameter $b$ here is related to the central charge.

As shown in [24, 45] the integral expression above can be obtained from the $\mathbb{S}^2 \times \mathbb{S}^1$ partition function of the $FM[SU(Q)]$ theory by taking a limit in which the $3d$ real mass parameters are scaled with the $\mathbb{S}^1$ radius. The sum over fluxes plus contour integrals are then traded for an integral in the complex plane.

The kernel function $K_{\Delta}^{(Q)}(x_1, \cdots x_Q, y_1 \cdots y_Q)$ satisfies remarkable properties and it appears in various identities and manipulation of complex Dotesenko-Fateev integrals. In [24] these properties where reinterpreted as dualities for $3d$ $\mathcal{N} = 2$ theories.

# 6 Discussion

Let us briefly discuss our main findings and some open questions. In this paper we have defined a four dimensional model $E[USp(2Q)]$ which has several interesting properties. First, the dynamics of the model is rather intricate leading to symmetry enhancement in the IR. Second, combining $E[USp(2Q)]$ models together and studying RG flows one can deduce various dualities and connections to other well studied theories. Finally, the model appears as a building block for constructing four dimensional models obtained by reducing the rank $Q$ E-string on a torus with flux in the abelian subgroups of the $E_8$ global symmetry of the six dimensional theory. To obtain these four dimensional models we had to gauge the emergent symmetry thus making the models intrinsically strongly coupled.

The six dimensional rank $Q > 1$ E-string theory has $E_8 \times SU(2)_L$ symmetry and an interesting open question is to find models corresponding to compactifications with flux (also) in the Cartan of the $SU(2)_L$ symmetry. This goes beyond what we have discussed. Also it would be interesting to understand compactifications on general Riemann surface, as it was done for $Q = 1$ case [1, 71], and not just the torus.

Another interesting question is to relate the $E[USp(2Q)]$ model to domain wall theories in five dimensions. In quite a few examples by now [1, 16, 17, 21] the four dimensional theories corresponding to compactifications of six dimensional SCFTs on a cylinder with flux have been related to four dimensional domain wall theories in five dimensions. These domain wall models interpolate between effective five dimensional gauge theories obtained by reduction of six dimensional SCFTs on a circle with different values of holonomies for global symmetries. Moreover, as we have seen, the $E[USp(2Q)]$ theory by dimensional reduction and flows is related to (up to flip fields) the $T[SU(Q)]$ theory which is an S-duality domain wall in four dimensions [27]. It would be very interesting to understand better a systematics of derivation of such domain wall models. This would facilitate the derivation of four dimensional theories

for general flux compactifications on tubes, understanding which is lacking even in the simplest case of class $\mathcal{S}$. For an example of some recent progress on understanding domain walls, and other higher dimensional supersymmetric defects, in lower dimensions see [72].

It would also be interesting to further explore the connection between the Rains interpolation kernel $\mathcal{K}_c$ whose integral form coincides with the index of the $E[USp(2Q)]$ theory $\mathcal{I}_{E[USp(2Q)]}$ and the kernel function for complex DF integrals. One way to connect to the two would be to consider the lens index [73–75], or $\mathbb{S}^1 \times \mathbb{S}^3/\mathbb{Z}_p$ partition function, of the $E[USp(2Q)]$ theory $\mathcal{I}^{(p)}_{E[USp(2Q)]}$. The lens index reduces in the $p \to \infty$ limit to the 3d index $\mathbb{S}^2 \times \mathbb{S}^1$ [73]. In this way we could directly connect

$$\mathcal{I}^{(p)}_{E[USp(2Q)]} \xrightarrow[p\to\infty]{} \mathcal{Z}_{T[USp(2Q)]} \xrightarrow[real\,mass]{} \mathcal{Z}_{FM[U(Q)]} \xrightarrow[2d\,limit]{} K^{(Q)}_\Delta \,.$$

This suggests the existence of a lens generalization of the interpolation functions of [32] from which the elliptic kernel was derived.

# Acknowledgements

We would like to thank Noppadol Mekareeya for valuable discussions. SP and SSR would like to thank the organizers of the Pollica summer workshop for hospitality during the final stages of this project. The workshop was funded in part by the Simons Foundation (Simons Collaboration on the Non-perturbative Bootstrap) and in part by the INFN, and the authors are grateful for this support. SP is partially supported by the ERC-STG grant 637844-HBQFTNCER and by the INFN. SSR is supported by Israel Science Foundation under grant no. 2289/18 and by I-CORE Program of the Planning and Budgeting Committee. M.S. is partially supported by the ERC-STG grant 637844-HBQFTNCER, by the University of Milano-Bicocca grant 2016-ATESP-0586 and by the INFN. GZ is partially supported by World Premier International Research Center Initiative (WPI), MEXT, Japan.

# A  Flux basis and global symmetry

In this appendix we summarize various conventions and properties associated with fluxes. The fluxes are vectors in the root lattice of $E_8$, and we first need to choose a basis of Cartan generators to use for it. For this we use the $SO(16)$ subgroup of $E_8$ under which the adjoint of $E_8$ decomposes as,

$$\mathbf{248}_{E_8} \to \mathbf{120}_{SO(16)} + \mathbf{128}_{SO(16)} \,,$$

where the $\mathbf{120}_{SO(16)}$ is the adjoint of $SO(16)$ and the $\mathbf{128}_{SO(16)}$ is one of its chiral spinors.[10] We next choose to span the Cartan of $SO(16)$ in a basis such that:

$$\mathbf{16}_{SO(16)} = a_1 + \frac{1}{a_1} + a_2 + \frac{1}{a_2} + ... + a_8 + \frac{1}{a_8} \,, \tag{131}$$

where $a_i$ are the fugacities for the chosen Cartans. The flux is then given by the eight number $(n_1, n_2, ..., n_8)$, where $n_i$ is the flux in the Cartan associated with $a_i$. Different values of $n_i$ correspond to different fluxes with some notable exceptions. Specifically, fluxes related by Weyl transformations actually represent the same flux. Here, as we are using an $SO(16)$ basis,

---

[10]At the group level then the subgroup of $E_8$ is actually $Spin(16)/\mathbb{Z}_2$, where we mod by the center element acting non-trivially on the vector and the other chiral spinor. Nevertheless, we shall be cavalier about the group structure here and refer to it simply as $SO(16)$.

the Weyl symmetry of $SO(16)$ is explicitly manifest and is given by permutations of $n_i$'s and reflections, $n_i \rightarrow -n_i$, for any even number of $n_i$s. Note that reflections for odd number of $n_i$'s are an outer, rather than inner, automorphism and as the roots of $E_8$ contain the weight of a chiral spinor of $SO(16)$, vectors differing by this transformation describe different fluxes. Additionally we also have the $E_8$ Weyl group elements that are not contained in the $SO(16)$ Weyl group. These map some of the roots of $SO(16)$ to the weights of its spinor representation. We shall delay the explanation of how these act on the flux to later in this appendix.

As this basis is used to span the root space, we can also use it to write the various roots, and with some abuse of notations also the weights of various representations. This will be useful later when we discuss the symmetry preserved by the flux. First we consider the vector representation of $SO(16)$, which in this basis is given by $(\pm 1, 0, 0, 0, 0, 0, 0, 0)$+ permutations. The non-zero weights of the adjoint of $SO(16)$, which are the roots, are given by $(\pm 1, \pm 1, 0, 0, 0, 0, 0, 0)$ + permutations. Finally the weights of the spinor representations are $(\pm \frac{1}{2}, \pm \frac{1}{2}, \pm \frac{1}{2}, \pm \frac{1}{2}, \pm \frac{1}{2}, \pm \frac{1}{2}, \pm \frac{1}{2}, \pm \frac{1}{2})$, with even number of minus signs corresponding to one chirality and odd number to the other. The roots of $E_8$ in this basis then are given by the roots of $SO(16)$ plus the weights of one of its chiral spinors, which we shall choose to be the one with an even number of minus signs.

We next want to consider what is the symmetry preserved by a given flux. Specifically, the flux breaks the $E_8$ symmetry to a subgroup. For generic values this subgroup is just the Cartan of $E_8$, but for special values, it is possible to preserve more symmetry. We next describe how the subgroup preserved by a given flux can be determined.

The property of the preserved symmetry that we will use here is that its Weyl group fixes the chosen vector. Since Weyl groups are generated by reflections in the plane defined by the associated root vector, the Weyl element associated to a given root will fix the flux vector if and only if the flux vector is orthogonal to the associated root. Therefore, the roots of the preserved symmetry are the subset of all $E_8$ roots orthogonal to the flux vector.

It is convenient in these considerations to look at various subsets of roots and weights of the $E_8$ roots. Specifically we mentioned that the roots of the form $(\pm 1, \pm 1, 0, 0, 0, 0, 0, 0)$+ permutations build the $SO(16)$ subgroup of $E_8$. More generally, roots of the same form, but with $p$ terms forced to be zero build the $SO(16-2p)$ subgroup. We can also consider the roots of the form $(1, 1, 0, 0, 0, 0, 0, 0) + (-1, -1, 0, 0, 0, 0, 0, 0)$+ permutations or $(1, -1, 0, 0, 0, 0, 0, 0)$+ permutations, which build an $SU(8)$ subgroup of $SO(16)$. Similarly we can also build $SU(8-p)$ subgroups of $SO(16-2p)$.

We need also to consider how the spinor weights decompose in terms of these subgroups. Under the decomposition of $SO(16) \rightarrow SO(16-2p) \times SO(2p)$, the spinor decomposes to bispinors of the two groups. Under the decompositions of $SO(16) \rightarrow U(1) \times SU(8)$, the spinors decompose to all the rank $q$ antisymmetric representations, where for one chirality $q$ is even while for the other it is odd. In our case we have the spinor of the form $(\pm \frac{1}{2}, \pm \frac{1}{2}, \pm \frac{1}{2}, \pm \frac{1}{2}, \pm \frac{1}{2}, \pm \frac{1}{2}, \pm \frac{1}{2}, \pm \frac{1}{2})$ with even number of minus signs. Under the $SU(8)$ subgroup each group with a different number of minus signs form a different representation of $SU(8)$ [11]. Specifically, weights with $l$ minus signs span the rank $l$ antisymmetric representation of $SU(8)$. A similar statement also holds for the $SU(8-p)$ subgroups.

Finally, we want to consider some examples. First, consider the flux vector

$$\mathcal{F} = (1, 1, 0, 0, 0, 0, 0, 0).$$

The $SO(16)$ roots orthogonal to it are of the form $(1, -1, 0, 0, 0, 0, 0, 0)$, $(-1, 1, 0, 0, 0, 0, 0, 0)$ and $(0, 0, \pm 1, \pm 1, 0, 0, 0, 0)$ + permutations of the last six. The roots $(1, -1, 0, 0, 0, 0, 0, 0)$ and $(-1, 1, 0, 0, 0, 0, 0, 0)$ span an $SU(2)$ and the ones of the form $(0, 0, \pm 1, \pm 1, 0, 0, 0, 0)$

---

[11] The Weyl group of $SU(8)$ preserves the permutation symmetry of the $SO(16)$ Weyl group, but not the reflection symmetry. As a result weight related by reflections span different representations in $SU(8)$.

plus permutations of the last six entries span an $SO(12)$ so this flux breaks $SO(16)$ to $U(1) \times SU(2) \times SO(12)$. The spinor roots orthogonal to $\mathcal{F}$ are of the form $(\frac{1}{2}, -\frac{1}{2}, \pm\frac{1}{2}, \pm\frac{1}{2}, \pm\frac{1}{2}, \pm\frac{1}{2}, \pm\frac{1}{2}, \pm\frac{1}{2})$, $(-\frac{1}{2}, \frac{1}{2}, \pm\frac{1}{2}, \pm\frac{1}{2}, \pm\frac{1}{2}, \pm\frac{1}{2}, \pm\frac{1}{2}, \pm\frac{1}{2})$ + permutations of the last six, where the total number of minus signs is even. The last six terms span a chiral spinor of $SO(12)$ while the first two span the fundamental of the $SU(2)$. Therefore the preserved symmetry has a $U(1) \times SU(2) \times SO(12)$ subgroup with additional roots transforming in the $(\mathbf{2}_{SU(2)}, \mathbf{32}_{SO(12)})$. This span the root system of $U(1) \times E_7$, which is the preserved group.

As another example, consider the flux vector $(\frac{1}{2}, \frac{1}{2}, \frac{1}{2}, \frac{1}{2}, \frac{1}{2}, \frac{1}{2}, \frac{1}{2}, \frac{1}{2})$. From the $SO(16)$ roots only the ones of the form $(1, -1, 0, 0, 0, 0, 0, 0)$+permutations are orthogonal. These spans the adjoint of $SU(8)$ so this chosen flux breaks $SO(16)$ to $U(1) \times SU(8)$. From the spinor roots, only the ones of the form $(\frac{1}{2}, \frac{1}{2}, \frac{1}{2}, \frac{1}{2}, -\frac{1}{2}, -\frac{1}{2}, -\frac{1}{2}, -\frac{1}{2})$ + permutations are orthogonal. As there are four minus signs, these span the 4-index antisymmetric representation of $SU(8)$. The roots of the preserved symmetry then are those of $U(1) \times SU(8)$+ the weights of the 4-index antisymmetric representation of the $SU(8)$, which gives the roots of $U(1) \times E_7$.

We see that the two fluxes preserve the same symmetry. This is no coincidence as both are roots of $E_8$ and there is a Weyl element of $E_8$ that maps them to one another. This element is not contained in the Weyl group of the $SO(16)$ we use as a basis and we shall end this section by detailing its action. For this we return to the decomposition of $SO(16)$ to $U(1) \times SU(8)$ we used previously. We remind the reader that under that decomposition we have $\mathbf{120}_{SO(16)} \rightarrow \mathbf{63}^0_{SU(8)} + \mathbf{28}^2_{SU(8)} + \overline{\mathbf{28}}^{-2}_{SU(8)} + \mathbf{1}^0_{SU(8)}$ and $\mathbf{128}_{SO(16)} \rightarrow \mathbf{70}^0_{SU(8)} + \mathbf{28}^{-2}_{SU(8)} + \overline{\mathbf{28}}^2_{SU(8)} + \mathbf{1}^4_{SU(8)} + \mathbf{1}^{-4}_{SU(8)}$. An important thing to notice here is that while each representation is only invariant under the charge conjugation of both $SU(8)$ and $U(1)$, the combination of both is invariant under the charge conjugation of each individually. The former is an element of the Weyl group of $SO(16)$, but the latter describes an element of the Weyl group of $E_8$ that is not in the Weyl group of the chosen $SO(16)$ subgroup, as the combination of both representations give the decomposition of the fundamental representation of $E_8$.

An alternative way to see this is to embed $U(1) \times SU(8) \subset SU(2) \times E_7 \subset E_8$, where the $U(1)$ is the Cartan of the $SU(2)$ and the $SU(8)$ is a maximal subgroup of $E_7$. It is straightforward to show that the adjoint of $E_8$ decomposes as in the previous paragraph so this gives the same embedding of $U(1) \times SU(8)$. The charge conjugation of the $SU(8)$ is part of the Weyl group of $E_7$, and the charge conjugation of the $U(1)$ is the Weyl group of $SU(2)$, and as these are in a direct product, these transformations can be done independently.

Having understood how this element acts, we can now use it to generate equivalent fluxes. For this we separate the flux part in the $U(1)$ and $SU(8)$ parts and reflect the latter. The $U(1)$ is spanned by the vector $(1, 1, 1, 1, 1, 1, 1, 1)$, and the remaining seven linearly independent vectors span the Cartan of the $SU(8)$. As an example consider the flux vector $(1, 1, 0, 0, 0, 0, 0, 0)$, then we can implement this Weyl transformation as:

$$(1, 1, 0, 0, 0, 0, 0, 0) = \frac{1}{4}(1, 1, 1, 1, 1, 1, 1, 1) + \frac{1}{4}(3, 3, -1, -1, -1, -1, -1, -1) \rightarrow \quad (132)$$
$$\frac{1}{4}(1, 1, 1, 1, 1, 1, 1, 1) - \frac{1}{4}(3, 3, -1, -1, -1, -1, -1, -1) = (-\frac{1}{2}, -\frac{1}{2}, \frac{1}{2}, \frac{1}{2}, \frac{1}{2}, \frac{1}{2}, \frac{1}{2}, \frac{1}{2}).$$

So we see that indeed this element maps some of the roots of $SO(16)$ to the weights of its spinor representation. Overall, one can show that this element map the 28 $SO(16)$ roots of the form $(1, 1, 0, 0, 0, 0, 0, 0)$ + permutations to the 28 spinor weights of the form $(-\frac{1}{2}, -\frac{1}{2}, \frac{1}{2}, \frac{1}{2}, \frac{1}{2}, \frac{1}{2}, \frac{1}{2}, \frac{1}{2})$ + permutations, and similarly for the 28 opposite roots and 28 opposite spinor weights. The 56 $SO(16)$ roots of the form $(1, -1, 0, 0, 0, 0, 0, 0)$ + permutations and the 70 spinor weights of the form $(-\frac{1}{2}, -\frac{1}{2}, -\frac{1}{2}, -\frac{1}{2}, \frac{1}{2}, \frac{1}{2}, \frac{1}{2}, \frac{1}{2})$ + permutations are inside

the $SU(8)$ and so are mapped to minus themselves. Finally the two spinor weights of the form $\pm(\frac{1}{2},\frac{1}{2},\frac{1}{2},\frac{1}{2},\frac{1}{2},\frac{1}{2},\frac{1}{2},\frac{1}{2})$ are inside the $U(1)$ and so are invariant under this transformation.

We can combine this element with the Weyl elements of the chosen $SO(16)$ to generate many other elements of the $E_8$ Weyl group. By acting with these on chosen fluxes, it is possible to generate many equivalent fluxes.

# B   Supersymmetric index definitions

Let us summarize the basic notations used to compute the $\mathcal{N}=1$ superconformal index [28, 29]. For more comprehensive explanations and definitions see [31]. The index of a given SCFT in four space-time dimensions is a refined Witten index of the theory quantized on $\mathbb{S}^3 \times \mathbb{R}$,

$$\mathcal{I} = \text{Tr}(-1)^F e^{-\beta\delta} e^{-\mu_i \mathcal{M}_i}, \tag{133}$$

here $\delta = \frac{1}{2}\{\mathcal{Q}, \mathcal{Q}^\dagger\}$, with $\mathcal{Q}$ one of the Poincaré supercharges, and $\mathcal{Q}^\dagger = \mathcal{S}$ it's conjugate conformal supercharge, $\mathcal{M}_i$ are $\mathcal{Q}$-closed conserved charges and $\mu_i$ their associated chemical potentials. All the states contributing to the index with non vanishing weight have $\delta = 0$ which makes the index independent on $\beta$.

For $\mathcal{N} = 1$, the supercharges are $\{\mathcal{Q}_\alpha, \mathcal{S}^\alpha = \mathcal{Q}^{\dagger\alpha}, \widetilde{\mathcal{Q}}_{\dot\alpha}, \widetilde{\mathcal{S}}^{\dot\alpha} = \widetilde{\mathcal{Q}}^{\dagger\dot\alpha}\}$, with $\alpha = \pm$ and $\dot\alpha = \pm$ the respective $SU(2)_1$ and $SU(2)_2$ indices of the isometry group of $\mathbb{S}^3$ ($Spin(4) = SU(2)_1 \times SU(2)_2$). We choose without loss of generality $\mathcal{Q} = \widetilde{\mathcal{Q}}_-$ to define the index. With this particular choice it is common to define the index to depend on the following specific fugacities,

$$\mathcal{I}(p,q) = \text{Tr}(-1)^F p^{j_1 + j_2 + \frac{1}{2}r} q^{j_2 - j_1 + \frac{1}{2}r}, \tag{134}$$

where $p$ and $q$ are fugacities associated with the supersymmetry preserving squashing of the $\mathbb{S}^3$ [30]. $j_1$ and $j_2$ are the Cartan generators of $SU(2)_1$ and $SU(2)_2$, and $r$ is the generator of the $U(1)_r$ R-symmetry.

The index is computed by listing all gauge invariant operators one can construct from modes of the fields. The modes and operators are conventionally called "letters" and "words", respectively. The single-letter index for a vector multiplet and a chiral multiplet transforming in the $\mathcal{R}$ representation of the gauge×flavor group is

$$
\begin{aligned}
i_V(p,q,U) &= \frac{2pq - p - q}{(1-p)(1-q)}\chi_{adj}(U), \\
i_{\chi(r)}(p,q,U,V) &= \frac{(pq)^{\frac{1}{2}r}\chi_{\mathcal{R}}(U,V) - (pq)^{\frac{2-r}{2}}\chi_{\bar{\mathcal{R}}}(U,V)}{(1-p)(1-q)},
\end{aligned} \tag{135}
$$

where $\chi_{\mathcal{R}}(U,V)$ and $\chi_{\bar{\mathcal{R}}}(U,V)$ denote the characters of $\mathcal{R}$ and the conjugate representation $\bar{\mathcal{R}}$, with $U$ and $V$ gauge and flavor group matrices, respectively.

With the single letter indices at hand, we can write the full index by listing all the words and projecting them to gauge singlets by integrating over the Haar measure of the gauge group. This takes the general form

$$\mathcal{I}(p,q,V) = \int [dU]\prod_k PE[i_k(p,q,U,V)], \tag{136}$$

where $k$ labels the different multiplets in the theory, and $PE[i_k]$ is the plethystic exponent of the single-letter index of the $k$-th multiplet, responsible for listing all the words. The plethystic

exponent is defined by

$$PE[i_k(p,q,U,V)] = \exp\left\{\sum_{n=1}^{\infty} \frac{1}{n} i_k(p^n, q^n, U^n, V^n)\right\}. \tag{137}$$

Let us now specialize to the case of $USp(2N_c)$ gauge group. The full contribution for a chiral superfield in the fundamental representation of $USp(2N_c)$ with R-charge $r$ can be written in terms of elliptic gamma functions, as follows

$$
\begin{aligned}
PE[i_k(p,q,U)] &\equiv \prod_{i=1}^{N_c} \Gamma_e\left((pq)^{\frac{1}{2}r} z_i\right) \Gamma_e\left((pq)^{\frac{1}{2}r} z_i^{-1}\right), \\
\Gamma_e(z) \equiv \Gamma(z; p, q) &\equiv \prod_{n,m=0}^{\infty} \frac{1 - p^{n+1} q^{m+1}/z}{1 - p^n q^m z},
\end{aligned}
\tag{138}
$$

where $\{z_i\}$ with $i = 1, ..., N_c$ are the fugacities parameterizing the Cartan subalgebra of $USp(2N_c)$ and are the eigenvalues of the matrix $U$. In addition, in many occasions we will use the shorten notation

$$\Gamma_e\left(uz^{\pm n}\right) = \Gamma_e\left(uz^n\right) \Gamma_e\left(uz^{-n}\right). \tag{139}$$

In a similar manner we can write the full contribution of the vector multiplet in the adjoint of $USp(2N_c)$, together with the matching Haar measure and projection to gauge singlets as

$$\frac{\kappa^{N_c}}{2^{N_c} N_c!} \oint_{\mathbb{T}^{N_c}} \prod_{i=1}^{N_c} \frac{dz_i}{2\pi i z_i} \prod_{k<\ell}^{N_c} \frac{1}{\Gamma_e(z_k^{\pm 1} z_\ell^{\pm 1})} \prod_{k=1}^{N_c} \frac{1}{\Gamma_e(z_k^{\pm 2})} \cdots, \tag{140}$$

where the dots denote that it will be used in addition to the full matter multiplets transforming in representations of the gauge group. The integration is a contour integration over the maximal torus of the gauge group and $\kappa$ is the index of $U(1)$ free vector multiplet defined as

$$\kappa = (p; p)(q; q), \tag{141}$$

with

$$(a; b) = \prod_{n=0}^{\infty} (1 - a b^n) \tag{142}$$

is the q-Pochhammer symbol.

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
