# Peer review of "Rank $Q$ E-string on a torus with flux"

_SciPost Physics, doi:SciPost Phys. 8, 014 (2020)_

## Round 1 · Referee Report · Anonymous · 2019-12-29

Strengths

1- Very precised and detailed analysis of the $E[Usp(2Q)]$ theories (superpotential, spectrum, symmetry enhancement, computation of the anomalies, IR dualities and RG flow).
2- Connection of the above mentioned dualities to identities which recently appeared in the math literature, via the supersymmetric index.
3- Construction, based on the $E[Usp(2Q)]$ theories, of tubes with various $E_8$ fluxes, using repeated gluings. The gluings involve gauging certain global symmetries which only appear in the IR, making the 4d theories intrinsically strongly coupled.
4- Finally, closing the tubes, construction of a torus with fluxes which is identified with the compactification torus of the 6d E-string SCFT.

Weaknesses

The paper does not contain important weaknesses. What follows, in the Requested Changes section, is rather a list of remarks or typos.

Report

The paper is well written and contains beautiful physics nicely related to recent works in mathematics. The main focus of the paper is a specific 4d N=1 Lagrangian theory which flows in the IR to a theory with enhanced global symmetry, and which is the building block of a class of 4d compactifications on a torus with fluxes of the rank $Q$ E-string 6d SCFT.

Requested changes

1- I was not able to reproduce equation (4.18), and unless I made a mistake I think there might be typos in that equation (and punctuation signs should be removed).
2- In section 2, at the beginning of the 2nd paragraph, a "proper holonomy inside $E_8$" is mentioned, and then again at the beginning of section 2.2. It might be helpful to the reader to say more about this holonomy.
3- It might help the reader to add the fields $b$ to Figure 3.
4- Remind the reader what $p$ and $q$ refer to in equation (3.31), in particular in connection with (3.33). Also, in (3.33) the notation $\mathfrak{u}_i$ for the "log" of $u_i$ could be explicited.
5- Typos :
p.6 "The boundary conditions associated with this choice is as follows" -> are.
p. 45 "one of it's chiral spinors" -> its.
p. 45 "The weights of the adjoint of SO(16)" -> the non-zero weights.

  • validity: top
  • significance: high
  • originality: high
  • clarity: high
  • formatting: excellent
  • grammar: excellent

Author:  Shlomo Razamat  on 2020-01-14  [id 706]

(in reply to Report 1 on 2019-12-29)

We are grateful to the referee for their thoughtful and detailed comments and intend to implement the suggestions in the revised version.

We have rechecked the equation 4.18 and it is OK as written.
(One issue that often causes mistakes is that the antisymmetric fields should be remembered to be taken traceless.)

---

## Round 1 · Referee Report · Anonymous · 2020-1-6

Report

This paper studies the compactification to 4d of an important class of 6d N=(1,0) SCFTs, the E-string theories of arbitrary rank, generalizing recent work for the rank 1 case by Kim, Razamat, Vafa and Zafrir. The compactification is performed on a torus, for specific classes of fluxes that break the E8 factor of the 6d global symmetry algebra to a maximal subalgebra. For the rank-Q E-string theory, a suitable Lagrangian 4d N=1 theory dubbed E[USp(2Q)] is identified which can be used as building block to construct 4d theories that flow in the IR to the compactified 6d theories of interest. The agreement between these constructions is argued for convincingly by matching of anomalies.

The realization in terms of the E[USp(2Q)] theories provides a convenient way to compute superconformal indices, which in turn can be used to verify that the expected global symmetries of the compactified 6d theories are correctly reproduced in the IR. The paper also makes the interesting observation that the superconformal indices are closely related to the interpolation kernels which have been studied by E. Rains in a mathematical context. Various identities and limits of the interpolation kernels are related respectively to dualities and to RG flows between different 4d theories. Finally in the paper the reduction to three dimensions of the E[USp(2Q)] theories is also discussed, as well as their relation to well known 3d theories such as the T[SU(Q)] theory of Gaiotto and Witten.

The paper contributes meaningfully to the larger program of mapping out and understanding the properties of the 4d N=1 theories that can be obtained from compactification of 6d (1,0) SCFTs with fluxes. I recommend it for publication, provided that the following minor issues are addressed:
1- In equation 2.5, the U(1)USp(2Q)^2 anomaly appears to include a factor of Q which should not be there since the index of the fundamental representation of USp(2Q) is 1 for all Q.
2- The discussion in section 5 sometimes refers to the theory T[U(Q)] and sometimes to T[SU(Q)], and similarly for FM[U(Q)] and FM[SU(Q)], without spelling out how these theories are related to each other. It would be helpful to add a clarifying comment.
3- A typo in footnote 4 on page 9: "N = 1" should be "n = 1".
4- In equation (3.13) for it would be helpful to the readers to specify how the missing (3,3) entry in the block matrix A_y is to be filled out.
5- The labels in equation (3.23) appear to be inconsistent with the previous text.
6- Equation (5.2) appears to be misplaced in the text, as it does not have a clear logical relationship to the preceding sentence.
7- In the sentence following equation (5.11), it appears that the correct reference should be to equation (5.5) and not (5.6).
8- The paper would benefit from further proofreading, as it contains a various typos as well as incomplete sentences which at times affect its readibility (for example: the last sentence on page 7 is difficult to parse; the last sentence on page 25 is incomplete; in the third sentence of the third paragraph of page 40, ". So that the" should be replaced by ". The").

---

## Round 2 · Author Response

We are grateful to the referees for their thoughtful and detailed comments. We have implemented the suggestions of both referees.

---

## Round 2 · List of Changes

Referee I:

1 -- We have rechecked the equation 4.18 and it is OK as written. (One issue that often causes mistakes is that the antisymmetric fields should be remembered to be taken traceless.)

2 -- We have added footnote 1 on page 4 to comment on the needed holonomy.

3 -- We have added the reference to fields $b_n$ in the caption of Figure 3.

4 -- We added footnote 8 to refer to equation 3.19 for the proper relation between the parameters.

We have fixed the typos found by the referee.

Referee II:

1 -- We fixed equation 2.5. We thank the referee for spotting this typo.

2 -- We streamlined the references to different types of models. In particular we now refer only to $FM[SU(Q)]$ and not to $FM[U(Q)]$. The partition function of FM[U(Q)] appears only in intermediate steps of the calculation and we added a comment around 5.24 to explain how it is related to the one of $FM[SU(Q)]$.

4 -- We added a comment below 3.13 explaining how the entry should be determined.

6 -- We have rephrased the discussion around equation 5.2.

We have fixed the typos and parsing issues mentioned by the referee.

---

## Editorial Decision

published